# Therapeutic potential of the secreted Kazal-type serine protease inhibitor SPINK4 in colitis

Ying Wang[1,8], Jing Han[1,2,8], Guang Yang[3,8], Shuhui Zheng[4,8], Gaoshi Zhou[1,8], Xinjuan Liu[5,8], Xiaocang Cao[6,8], Guang Li[5], Bowen Zhang[7], Zhuo Xie[1], Li Li[1], Mudan Zhang[1], Xiaoling Li[1], Minhu Chen[1] & Shenghong Zhang[1,2]✉

Mucus injury associated with goblet cell (GC) depletion constitutes an early event in inflammatory bowel disease (IBD). Using single-cell sequencing to detect critical events in mucus dysfunction, we discover that the Kazal-type serine protease inhibitor *SPINK4* is dynamically regulated in colitic intestine in parallel with disease activities. Under chemically induced colitic conditions, the grim status in *Spink4*-conditional knockout mice is successfully rescued by recombinant murine SPINK4. Notably, its therapeutic potential is synergistic with existing TNF-α inhibitor infliximab in colitis treatment. Mechanistically, SPINK4 promotes GC differentiation using a Kazal-like motif to modulate EGFR-Wnt/β-catenin and -Hippo pathways. Microbiota-derived diacylated lipoprotein Pam2CSK4 triggers SPINK4 production. We also show that monitoring SPINK4 in circulation is a reliable noninvasive technique to distinguish IBD patients from healthy controls and assess disease activity. Thus, SPINK4 serves as a serologic biomarker of IBD and has therapeutic potential for colitis via intrinsic EGFR activation in intestinal homeostasis.

Inflammatory bowel disease (IBD), mainly including ulcerative colitis (UC) and Crohn's disease (CD), is characterized by chronic and recurrent inflammation distributed throughout the digestive tract and has an undetermined etiology[1,2]. The mucus layer is the first line of intrinsic defense, as it spatially segregates commensal microbiota from the host[3]. Recently, a study focused on mucus damage in IBD and defined it as an early incident during the genesis and development of UC[4]. The mucus layer was quantitatively and structurally weakened owing to goblet cell (GC) dysfunction in a dextran sodium sulfate (DSS)-induced colitis model, followed by recovery before colitis remission[5]. Conversely, interleukin 10-deficient mice had a penetrable colonic mucus layer with increased thickness compared with that of DSS-induced mice[6]. GCs, which maintain mucus integrity stimulated by cytokines, neurotransmitters, or microbiota[7], play a pivotal role in colitis. The core colonic mucin, mucin-2 (MUC2), is necessary for developing the mucus layer[8], and spontaneous intestinal inflammation is observed in MUC2-knockout (KO) mice[9,10]. A defect in exocytosis or the rapid degradation of MUC2 is also an early indicator of the increased susceptibility associated with colitis[11,12]. Furthermore, mucin glycosylation or dietary fibre-deprived mucolytic microbiota could spontaneously induce colitis[11,13], whereas mucus sialylation affects host-commensal homeostasis[14].

[1]Division of Gastroenterology, The First Affiliated Hospital, Sun Yat-sen University, Guangzhou, P. R. China. [2]Division of Gastroenterology, Guangxi Hospital Division of The First Affiliated Hospital, Sun Yat-sen University, Nanning, P. R. China. [3]Department of Minimally Invasive Intervention, State Key Laboratory of Oncology in South China, Guangdong Provincial Clinical Research Center for Cancer, Sun Yat-sen University Cancer Center, Guangzhou, P. R. China. [4]Research Center for Translational Medicine, The First Affiliated Hospital, Sun Yat-sen University, Guangzhou, P. R. China. [5]Department of Gastroenterology, Beijing Chaoyang Hospital, Capital Medical University, Chaoyang District, Beijing, P. R. China. [6]Department of Hepato-Gastroenterology, Tianjin Medical University General Hospital, Tianjin Medical University, Tianjin, P. R. China. [7]College of Life Sciences, Beijing Normal University, Beijing, P. R. China. [8]These authors contributed equally: Ying Wang, Jing Han, Guang Yang, Shuhui Zheng, Gaoshi Zhou, Xinjuan Liu, Xiaocang Cao. ✉e-mail: zhshh3@mail.sysu.edu.cn

Despite these findings, the role of colonic GCs, typically characterized by mucin production, in colitis has been underestimated[15]. Several studies have uncovered the novel functions of GCs in targeting congenital immunity following the emergence of single-cell RNA-sequencing (scRNA-seq)[16,17]. Mucins produced by sentinel GCs possess immunological characteristics, and their production is triggered by NLR Family Pyrin Domain Containing 6 (Nlrp6)[18]. Considering the dysfunction of GCs in IBD[19], their generation and differentiation are increasingly important for therapy development for colitis. Although the essential roles of the epidermal growth factor receptor (EGFR), Wnt/β-catenin, Notch, and Hippo pathways for epithelium regeneration have been well investigated, data on the differentiation of GCs in IBD, which affects the mucus recovery in colitis, remain limited[20,21].

To detect the critical factors underlying mucus dysfunction in IBD, we performed single-cell analysis and discovered that serine protease inhibitor Kazal-type 4 (*SPINK4*), a noteless participant in the serine protease inhibitor family, was dynamically regulated in colitic intestine. This family possesses a structure similar to that of epidermal growth factor (EGF) and consists of secretory proteins that share the Kazal domain of serine protease inhibitors[22]. The most recognized member of the SPINK family is SPINK1, which contributes to chronic pancreatitis and cancers[23,24]. SPINK6 participates in the repair of the epidermal barrier[25] and promotes cell migration and invasion in nasopharyngeal carcinoma[22]. SPINK4 is sightly expressed in intestinal GCs specifically located in the digestive tract under normal conditions; however, the underlying mechanism remains nebulous[26,27]. There are few studies on SPINK4 that uncover its predictive value in colorectal cancer[28].

Therefore, the aim of this study is to investigate the role and underlying mechanisms of SPINK4 in colitis. Our study provides novel insights into the therapeutic potential of SPINK4 in IBD.

## Results

### SPINK4 is dynamically regulated in IBD patients and colitis model and specifically localized to GCs

To excavate the critical factors connecting GC dysfunction to IBD, we performed scRNA-seq based on endoscopic tissues from UC patients. In line with the RNA-sequencing (RNA-seq, 38 cases) data in our center (Supplementary Fig. 1a) and mucosal profiling from pediatric IBD[27], *SPINK4* was identified as a novel marker of GCs with significantly elevated levels in the inflamed region from UC patients compared with those in healthy individuals or normal parts from the patients (Fig. 1a, Supplementary Fig. 1b).

We further examined *SPINK4* expression in a validated group and found that it was highly upregulated in endoscopic specimens from the IBD patients (Fig. 1b, c). The abundance of *SPINK4* was also confirmed in colonic samples from surgery, with higher expression in inflammatory lesions than that in relative normal tissues (Supplementary Fig. 2a). Further, *SPINK4* was enriched and located in the mucosal layer in IBD colonic tissues compared with surgical controls (Supplementary Fig. 2b).

We also assessed the *Spink4* expression in acute and chronic colitis models induced by DSS or 2,4,6-trinitrobenzene sulfonic acid (TNBS). Interestingly, *Spink4* was dynamically regulated in the inflamed mucosa. The abundance of *Spink4* was higher in colitis models than in the controls (Fig. 1d and Supplementary Fig. 2c, d); however, *Spink4* was barely expressed in the active stage and upregulated during remission in the DSS-induced colitis intestine with a significant difference, consistent with GC deficiency (Fig. 1e).

Consistent with our mRNA data, SPINK4 protein was highly expressed in intestinal specimens with mild inflammation in TNBS models and dramatically elevated in DSS sections during the remission stage (Fig. 1f, h and Supplementary Fig. 2e). Considering the dynamic progression of TNBS models, we assessed SPINK4 expression at distinct developmental periods (e.g., days 2, 5, 8 post-enema). SPINK4

expression in TNBS models was relatively low before the active stage, in parallel with reduced mucus layer thickness and injured GCs, increased within overwhelming inflammation, and finally reversed to the baseline level under remission (Fig. 1g, h, Supplementary Fig. 2f).

As illustrated by our scRNA-seq analysis, SPINK4 specifically originated from GC rather than GC progenitor cells or other epithelial parts (Fig. 1a). Furthermore, the distribution of SPINK4 was overlapped with MUC2, especially in the middle and bottom of the intestinal crypt in IBD samples (Fig. 1i). We also validated *SPINK4* mRNA localization to verify the origination (Fig. 1j, Supplementary Fig. 2g). On the account of pseudotime order, it burst from an early subset of the population, which was distinguished from the mature GC at the terminal stage (Fig. 1a, Supplementary Fig. 1c, d). Henceforth, SPINK4 exerted a significant effect as a secreted form from the GC population initially.

### Recombinant SPINK4 protein (rSPINK4) rescues colitis by maintaining intestinal homeostasis

To examine the potential beneficial effect of SPINK4 in IBD, we constructed two colitis animal models and delivered murine rSPINK4 by intraperitoneal injection (Fig. 2a). We noticed that the body weight of TNBS-induced colitis mice rapidly increased after treatment with rSPINK4 (Fig. 2b). This group exhibited a slightly extended colon length (Fig. 2c, Supplementary Fig. 3a), marginally reduced intestinal permeability (Supplementary Fig. 3b), and lower inflammation activity (Fig. 2d, Supplementary Fig. 3c). Considering the specific localization of SPINK4, we further tested GCs numbers and mucus thickness and found that both values in the rSPINK4 treatment group were increased, whereas the tissues from rSPINK4 group had a lower proliferation potency (Supplementary Fig. 3d). Consistently, transmission electron microscopy (TEM) test analysis revealed that more mature vesicles accumulated in GCs in the rSPINK4 treatment group (Supplementary Fig. 3d). Hence, both intestinal epithelial cells (IECs) and GCs are essential for intestinal homeostasis during the process of colitis recovery.

To clarify the protective effects of endogenous SPINK4 on GCs, we established *Spink4*-conditional knockout (cKO) mice. Immunohistochemical evaluation revealed a complete absence of SPINK4 in the intestinal epithelium of cKO mice (Supplementary Fig. 3h). No significant difference was observed in the colonic length and body weight between cKO and wild-type (WT) mice under normal conditions (Fig. 2e–g). Additionally, there was no evidence of mucosal erosion or neutrophil infiltration in the colonic tissue of cKO mice upon digital and histological evaluation (Fig. 2k, Supplementary Fig. 3h, i). After DSS administration, cKO mice showed a more dramatic body weight loss (Fig. 2e) and increased disease activity (Supplementary Fig. 3e) than co-housed controls after DSS administration. More shortening colon was observed in the cKO group than in the control group after DSS administration (Fig. 2f, g). Increased intestinal permeability (Fig. 2h), in addition to severe intestinal inflammation and mucosal damage (Fig. 2i, Supplementary Fig. 3f, g), emerged in the cKO group. GCs and the mucus layer showed a contrary tendency with SPINK4 deletion, which is indicative of mucus dysfunction (Fig. 2i, j). In addition, the secretory vesicles in GCs appeared to be immature in cKO mice (Fig. 2i), and the secretory cell biomarker of Paneth and enteroendocrine cells were seldomly affected by SPINK4 deficiency (Fig. 2i, j). However, the proliferation potency in cKO colitis mice was diminished even without DSS treatment (Supplementary Fig. 3i), consistent with the slightly decreased levels of ISC RNA markers (Supplementary Fig. 3j), which might be due to the balance required for intestinal homeostasis supported by endogenous SPINK4. Female mice administered with DSS (Supplementary Fig. 3k–n) and the TNBS-induced cKO colitis mice (Supplementary Fig. 4a–e) showed similar performance to that of the DSS-induced male ones mentioned previously.

To further verify the directed differentiation induced by SPINK4, we established intestinal organoids from cKO mice and co-housed

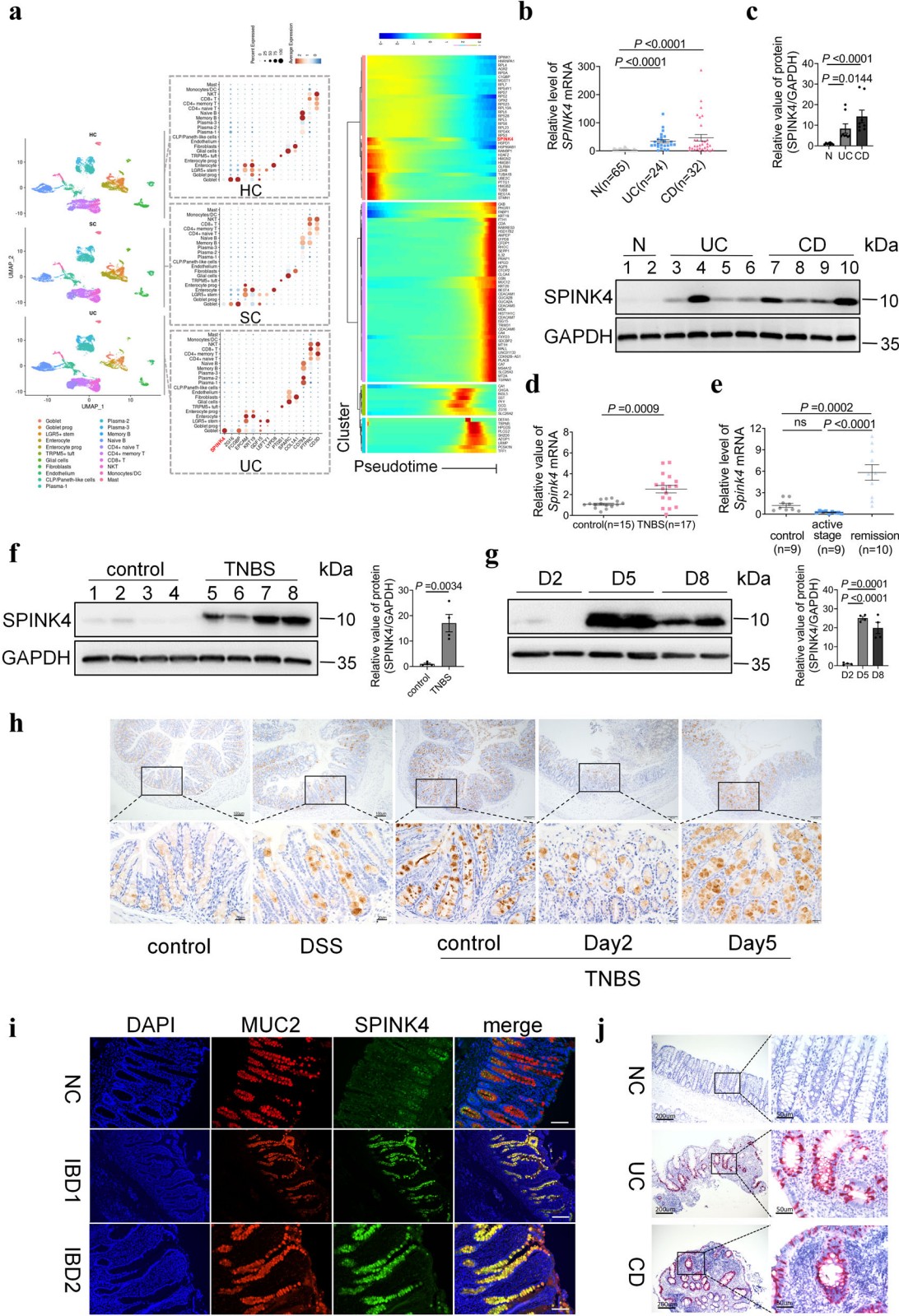

controls and observed no difference between them in the number of GCs under normal conditions, whereas the expression of MUC2 was distinctly diminished in the cKO group within increased concentrations of TNF-α (Supplementary Fig. 4f). After DSS induction, cKO colitis mice treated with the rSPINK4 protein consistently exhibited reduced inflammation infiltration and weight loss accompanied by extended colon length, which represented colitis recovery (Fig. 2k–m,

Supplementary Fig. 4g) compared with the untreated mice. In luminol-based in vivo imaging, the cKO group showed a higher incidence of ulcers, erosions, and mucosal ischemia than the wild-type mice and the rSPINK4 treatment group (Fig. 2k). Furthermore, the inflammation activity and intestinal permeability were in line with the forecast (Supplementary Fig. 4h–k). The MUC2 expression, mucus thickness, and the proliferation index in cKO mice treated with rSPINK4 were also

**Fig. 1 | Upregulation of SPINK4 in the intestines of IBD patients and colitis mice. a** Single-cell profiling based on colonic biopsies. Left: uniform manifold approximation and projection (UMAP) plot in healthy controls (HC), uninflamed regions from patients (SC), and inflamed regions from patients (UC); middle: dot heatmap presents the gene expression in each group. Each row indicates a different cell type, while each column represents an individual gene. The color depth of each circle is correlated with the level of gene expression, and the size corresponds with the proportion of cells; right: heatmap of the top 100 genes under pseudotime condition presented in the UC group. **b** qRT-PCR results of relative *SPINK4* level in the intestinal mucosa (CD, $n = 32$; UC, $n = 24$; N, $n = 65$). **c** Representative SPINK4 level in endoscopic specimens via Western blotting. Lines 1–2 represent healthy controls; lines 3–6 and lines 7–10 represent UC and CD patients, respectively. **d, e** Relative level of *Spink4* in the TNBS model (**d**) at the active stage and DSS models (**e**) at the active and remission stages. **f, g** Differential expression of SPINK4 in TNBS-induced colitis models at the active stage (**f**) and the whole stage (**g**) via immunoblotting. Left panel: controls (lines 1–4) and TNBS administration (lines 5–8); Right panel: SPINK4 expression in injured intestines including day 2 (D2), 5 (D5), and 8 (D8) after TNBS enema. **h** Intestinal staining targeting SPINK4 from two models at different stages; Day 2 and Day 5 refer to the time after TNBS enema. **i** Double immunofluorescence staining targeting SPINK4 and MUC2 in healthy control (NC) tissue and inflammatory sections of IBD with DAPI counterstaining. Scale bar: 100 μm. **j** RNA localization of *SPINK4* in intestinal mucosa from CD and UC patients, and negative controls (NC) showed. Red dots denote the mRNA localization of *SPINK4* with hematoxylin counterstaining. Data are presented as the mean ± SEM. All tests were two-sided. Statistical significance was calculated using unpaired Student's *t* test (**d, f**) and one-way analysis of variance (ANOVA) (**e, g**). Kruskal–Wallis test was performed in non-normal data (**b, c**); $n = 2$–4 biologically independent experiments (**c, f, g**). Source data are provided as a Source Data file.

in accordance with those of WT mice (Supplementary Fig. 4l). We also assessed the expression of secretory lineage biomarkers and inflammatory factors, apparent reverse came up with the level of MUC2, IL-6, and TNF-α upon rSPINK4 served (Supplementary Fig. 5a–c).

Infliximab, a TNF-α inhibitor, is used to treat moderate to severe CD[29]. Remarkably, the rSPINK4-infliximab recipe almost eliminated macroscopic inflammation compared with that in the control group, exhibiting a synergistic effect (Fig. 2n, Supplementary Fig. 4m, n). Based on dual drug therapy, histological assessment revealed a significant reduction in erosion and ulceration range as well as decreased inflammation infiltration (Fig. 2o, Supplementary Fig. 4o).

## SPINK4 induces the differentiation of ISCs into GCs through the EGFR pathway

We explored the underlying mechanisms of SPINK4 by establishing mouse intestinal organoids and measuring the IEC markers under rSPINK4 stimulation. We observed a high expression of the secretory lineage markers, Tuft cells, and enteroendocrine cells, along with a slight elevation in those of absorptive cells in mouse organoid (Fig. 3a). These data suggest that SPINK4 contributes to colitis recovery through epithelium reconstruction.

SPINK family members share a similar Kazal motif like EGF[22]. Therefore, we examined whether SPINK4 would function through the EGFR signalling pathway. The dose-dependent effect of SPINK4 on phosphorylated EGFR (Y1608) was measured in the intestinal cell line NCM460 and colonic cancer cell line HT-29 (Fig. 3b). Furthermore, the Ras-MAPK and AKT pathways, directly downstream of the EGFR pathway, were also significantly activated (Fig. 3c). This influence exerted by rSPINK4 could be abolished by AG1478, a tyrosine kinase inhibitor (Supplementary Fig. 5d). In addition, it was found that the downstream processing was partially obstructed when EGFR was knocked out in GCs (Supplementary Fig. 5e). However, a few changes in GCs were present in the organoids established from cKO and WT mice, which made it difficult to determine whether rSPINK4 participates in these pathways under normal conditions (data not supplied). AG1478 induced apoptosis, which did not require SPINK4 function (Supplementary Fig. 5f). The number of GCs slightly increased, and the organoid morphology changed into a cavity-type structure with thick walls following rSPINK4 treatment under inflammatory conditions without EGF stimulation (Fig. 3d, Supplementary Fig. 5f). This effect was abolished by AG1478, and intrinsic SPINK4 also contributed to the GC homeostasis following TNF-α treatment (Fig. 3d, Supplementary Fig. 5f). Except for GCs, the proliferation of epithelia and the number of endocrine cells were slightly decreased in the cKO group, regardless of the presence of TNF-α. The number of Ki67-positive cells was also reduced after rSPINK4 administration and decreased further following AG1478 treatment in the inflammatory environment, suggesting that this process may be independent of EGFR (Supplementary Fig. 5f). We observed similar results in CD patients, indicating that SPINK4 participated in not only differentiation but also proliferation in vitro

(Supplementary Fig. 6a). rSPINK4 could also induce the proliferation of NCM460 and Caco-2 cells in normal conditions, whereas cell proliferation promoted by rSPINK4 was reversed by TNF-α (Supplementary Fig. 6b, c). These results indicated that SPINK4 participated in the normal regeneration of the epithelium, and the differentiation process was mediated through the EGFR pathway in response to the shortage of GCs under a stimulated environment.

As SPINK4 functions via the EGFR pathway, we conducted an in vivo test using a tyrosine kinase inhibitor, gefitinib. The gefitinib group exhibited a significant reduction in weight loss and colon length, with an altered vascular pattern and severe mucosal erosion, consistent with our findings using AG1478. This indicates an accumulation of colitis activity in gefitinib administration group based on rSPINK4 treatment (Fig. 3e–g, Supplementary Fig. 6d, e). Within SPINK4 deficiency in the epithelium, the expression of phosphorylated EGFR in colitis lesions decreased (Fig. 3h). rSPINK4 mitigated the shortage of activation of the EGFR pathway (Fig. 3h). We also validated EGFR phosphorylation in the inflammatory tissues (Supplementary Fig. 6f). On account of the crucial role of EGFR in intestinal epithelial regeneration, we conducted an assessment of other potential ligands for EGFR; however, no other differentially expressed ligands originating from the epithelia were identified in patients with IBD (Supplementary Fig. 6g). These results suggest that SPINK4, one of the few upstream molecules of EGFR, plays a protective role in the pathology of IBD.

## SPINK4 regulates the Hippo and Wnt/β-catenin pathways by directly binding to specific subdomains of EGFR

Considering the similar structure to EGF, we further examined if rSPINK4 rescued colitis by directly interacting with EGFR. We detected an interaction between EGFR and SPINK4- FLAG after the ex-vivo incubation in concentrated medium and cell extraction from IECs overexpressing SPINK4-FLAG (Fig. 4a). Accordingly, the in vitro incubation of rSPINK4 and cell extraction confirmed the ability of EGFR to interact with rSPINK4. (Fig. 4b). The conjugation between rSPINK4 and the recombinant extracellular domain of EGFR was validated using a pull-down assay (Fig. 4c). To investigate the native localization of SPINK4, IECs were stimulated with rSPINK4, which was labeled with a fluorescent dye. Surprisingly, conjugated SPINK4 was detected by confocal laser scanning after 2 min and partially overlapped with EGFR. Conjugated SPINK4 was endocytosed, and a diminishing fluorescence signal was obtained over time. The apparent formation of vesicles or clusters of EGFR was displayed in the cytoplasm within minutes, which was dominantly localized in the plasma membrane originally (Fig. 4d). We also examined the ligand function of SPINK4 using an isothermal titration calorimetry (ITC) assay. Only a small amount of heat was released during the titration of the rSPINK4 solution (Fig. 4e), indicating a weak interaction between SPINK4 and EGFR.

Additionally, a competitive immunoblot assay revealed that SPINK4 could compete with EGF for EGFR binding in a dose-dependent

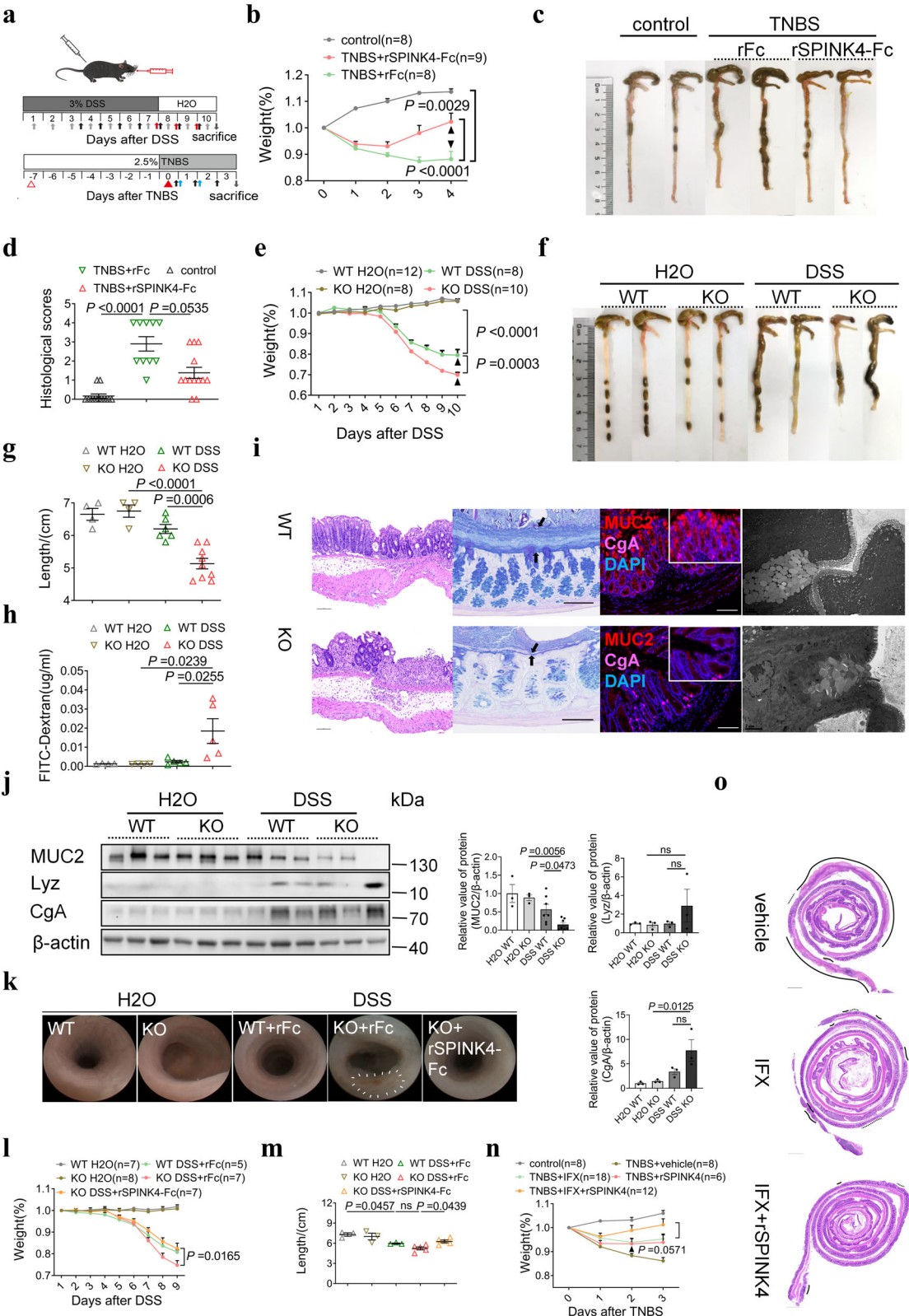

manner. Moreover, the binding effects of SPINK4 to EGFR could be quantitatively estimated using an EGFR binding test (Fig. 4f, g).

According to the function of the extracellular domain of EGFR[22], we identified that the basic subdomain of EGFR interacting with SPINK4 was the 361–637 amino acid sequence known as the receptor L-domain and growth factor receptor domain 4 (Fig. 4h). We found that the enzyme site and disulfide residues were indispensable for the binding reaction between SPINK4 and EGFR (Fig. 4i). To some extent,

the predictive interactive model between SPINK4 and EGFR (Fig. 4j) was in accordance with the results mentioned before (Fig. 4h, i).

We then examined the downstream targets of the SPINK4-EGFR pathway and identified several key points, such as the Wnt/β-catenin, Notch, BMP and Hippo pathways, which are essential for epithelium generation[20,21]. Based on the evaluation of pivotal molecules, we observed a dose-dependent activation of the Wnt/β-catenin pathway, whereas the Hippo pathway was inhibited (Supplementary Fig. 7a); the

**Fig. 2 | SPINK4 alleviates colitis inflammation and induces GC regeneration.**
**a** Schematic overview of in vivo experiment; the black upwards arrow indicates
rSPINK4, red indicates gefitinib, blue stands for infliximab and gray represents
Pam2CSK4. The red triangles represent the pre-sensitization and enema with TNBS.
The downwards arrow indicates sacrifice time. **b**–**d** Weight loss (**b**), representative
colonic images (**c**) and histological scores (**d**) in the ethanol group (control, $n = 8$),
TNBS group (TNBS + rFc, $n = 8$), and rSPINK4-Fc treatment group (TNBS + rSPINK4-
Fc, $n = 9$). **e** Weight loss in WT $H_2O$ ($n = 12$), KO $H_2O$ ($n = 8$), WT mice administered
DSS solution (WT DSS, $n = 8$), and cKO mice administered DSS solution (KO DSS,
$n = 10$). **f**, **g** Appearance of colon (**f**) and intestinal length (**g**) in the four groups.
**h** Serum level of fluorescence signal at excitation/emission wavelength of 490/
530 nm. **i** Typical images from cKO and co-housed colitis mice presented with H&E,
AB/PAS staining (arrow points to mucus layer), double immunofluorescence for
MUC2 (red), CgA (pink) counterstained with DAPI dye (blue), and TEM test. Unla-
beled scale bar: 100 μm. **j** Secretory biomarkers for GCs (MUC2), Paneth cells (Lyz),

and enteroendocrine cell (CgA) in the four groups. **k** Representative pictures of the
luminol-based in vivo endoscopic imaging (the circle represents the erosion or
ulceration) in WT $H_2O$ ($n = 7$), KO $H_2O$ ($n = 8$), WT mice administrated DSS and rFc
(WT DSS+rFc, $n = 5$), cKO mice administrated DSS and rFc (KO DSS+rFc, $n = 7$), and
cKO mice with rSPINK4-Fc treatment (KO DSS+rSPINK4-Fc, $n = 7$). **l**, **m** Weight loss
(**l**) and length changes (**m**) in the five groups. **n** Weight change in infliximab (IFX,
$n = 18$), rSPINK4 ($n = 6$), and combined treatment ($n = 12$) compared with TNBS
models (TNBS + vehicle, $n = 8$) and control group ($n = 8$). **o** Representative H&E
image of Swiss roll intestine. The dotted line represents the inflammatory cell
infiltration, while the continuous line describes severe ulceration. Data are pre-
sented as the mean ± SEM. All tests were two-sided. Statistical significance was
calculated using unpaired Student's $t$ test (**n**) and one-way ANOVA (**b**, **e**, **g**, **h**, **j**, **l**, **m**).
Kruskal–Wallis test was performed in non-normal data (**d**); $n = 3$–13 biologically
independent experiments (**d**, **h**, **j**, **l**, **m**). Source data are provided as a Source
Data file.

---

effects were reversed by the administration of AG1478 or the deletion of
EGFR (Supplementary Fig. 7b, c). As expected, the Hippo pathway was
downregulated, while the Wnt pathway was upregulated in colitis,
especially during the transitional period of recovery (Supplemen-
tary Fig. 7d).

### Endogenous SPINK4 is triggered by Pam2CSK4 and affects the property of the mucus layer

Given the fact that SPINK4 acts as a protective factor against intestinal
attack, we examined the origin of SPINK4 in our models. A minimal
difference in SPINK4 expression was observed in IECs under inflam-
matory conditions (Supplementary Fig. 8a). Birchenough et al.[18]
revealed that MUC2 expression is positively influenced by pathogenic
factors, except for proinflammatory cytokines. Therefore, we co-
cultured LS174T cells, a colonic cancer cell line, which mimics GCs
in vitro on account of abundant mucous vesicles, with microbiotic
elements[18,30,31]. Pam2CSK4, a synthetic diacylated lipopeptide that
activates the TLR2/TLR6 heterodimer[32], elevated the SPINK4 level in
the supernatant from LS174T (Fig. 5a). As the positive control of
MUC2 stimulation, carbachol also acted as an efficient promoter of
SPINK4 expression through the parasympathetic nervous system in a
dose-dependent manner (Fig. 5b). The other microbiotic
components[18] had no effect on SPINK4 secretion (Supplementary
Fig. 8b–f). To further validate the function, we examined SPINK4
expression after intracellular extraction. *SPINK4* levels notably
increased in organoids that originated from patients and WT mice after
being stimulated with Pam2CSK4 (Fig. 5c, d). The intracellular extrac-
tion from cKO mice showed no effect on SPINK4 expression, along
with little influence on MUC2 levels at a high concentration, whereas
Pam2CSK4 contributed to elevated levels of SPINK4 and MUC2 in co-
housed controls (Fig. 5e). Additionally, in vivo, the Pam2CSK4 treat-
ment group exhibited increased body weight and colonic length under
colitic attack compared to PBS treatment group after DSS adminis-
tration (Fig. 2a, Supplementary Fig. 8g, h). Further histological staining
was performed, which demonstrated that Pam2CSK4 can alleviate
colitis by triggering SPINK4 production and GC remodeling (Supple-
mentary Fig. 8i). Taken together, the stimulation of SPINK4 production
by bacteria-derived Pam2CSK4 promotes epithelial regeneration,
particularly the regeneration of GCs during the early stage of colitis.

We also assessed the intracellular influence of SPINK4 on GCs.
Firstly, the gel-forming mucins, rather than transmembrane mucins,
were depleted in *SPINK4*-KO cells (Fig. 5f). Moreover, MUC2 levels were
rewired along with SPINK4 and phosphorylated EGFR levels in *SPINK4*-
KO and *SPINK4*-overexpressing cells (Fig. 5g), which indicated that
SPINK4 may participate in mucin production through the regulation of
EGFR. MUC2 levels in endoscopic samples and the strong correlation
between *SPINK4* and *MUC2* support this hypothesis (Fig. 5h, i). Finally,
we estimated the mucus permeability, which is the most important
intrinsic property of mucus. There was an increased permeability

associated with SPINK4 deficiency, not only in the colitis model but
also under normal conditions (Fig. 5j).

### Circulating SPINK4 originating from GCs serves as a distinct biomarker in IBD patients

Owing to its secretory characteristic, we hypothesized that SPINK4
exists in patients' circulation. Indeed, the SPINK4 levels were
approximately double in the sera of IBD patients, including 51 UC
patients and 108 CD patients, when compared with those in 64
healthy controls (Fig. 6a). Moreover, there was a mild difference
between the remission and active stages in CD patients with statis-
tical difference (Fig. 6b). Furthermore, SPINK4 performed with a
notably higher specificity for endoscopic evaluation than other
clinical indexes but not for clinical assessment (Fig. 6c, d, Supple-
mentary Table 1). We concluded that SPINK4 was more suitable as a
noninvasive marker that could partially substitute endoscopy.
Table 1 summarizes the basic information collected from volunteers;
a higher SPINK4 level was more common in L2 (33.0%) and L3 (35.6%)
than in L1 (Fig. 6e). This could be attributed to the different con-
centrations or characteristics of GCs between the small intestine and
colon. Therefore, we proposed a new index for distinguishing IBD
patients from healthy controls to assess disease activity instead of
invasive operation and to initially screen for early stages of CD
patients in L1, whose inflamed regions are confined to the small
intestine.

Our results suggest that SPINK4 participates in the regulation of
intestinal regeneration and differentiation by directly influencing the
EGFR pathway in colitis (Fig. 6f).

## Discussion
Only limited studies have provided a concrete mechanism of the
Kazal domain shared among serine protease inhibitor family
members[33,34]. Circulating SPINK4 is a valuable factor in the diagnosis
of colorectal carcinoma with no predictive value in survival[35]. The
ambiguous results of SPINK4 immuno-expression in colorectal tis-
sues with cancer involvement for outcome prediction are presented
in distinct studies[28,36]. Our group found that SPINK4, a GC secretory
factor at an early stage, played an essential role in the development
and progression of IBD and colitis models. It drove GC differentia-
tion by specifically binding and phosphorylating EGFR, which
modulated the downstream pathways. SPINK4 expression in the
intestine originates from a microbiotic component, especially in
gram-negative bacteria. Indeed, the variation in epithelial regen-
eration that relied on SPINK4 is a reactive process associated with
injury. When the mucus layer, the first line of defense, is attacked by
microbiota, the injured intestine is triggered to maintain this barrier
integrity via SPINK4 secretion. The inability of IBD to be self-healing
may be attributed to an insufficient amount of SPINK4 as well as
other pathogenetic mechanisms[37,38].

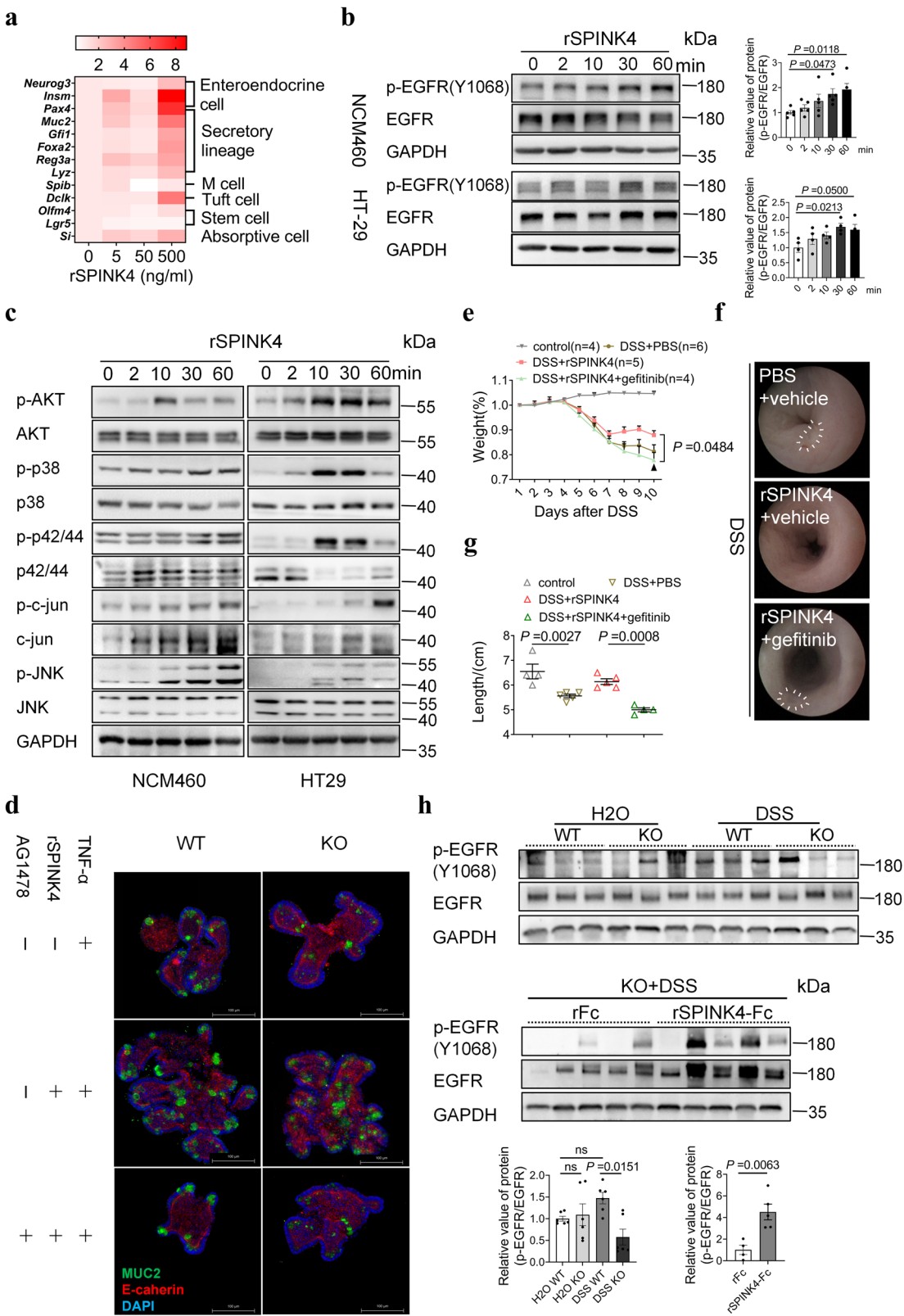

IBD has become a global burden among the digestive diseases[2]. Though numerous serological markers burst out[39,40], there remains a lack of efficient markers that could specifically identify IBD patients and entirely substitute for invasive examination in clinical evaluation. Fecal calprotectin is a remarkable factor in IBD management and reflects the intestinal inflammation in all colitis disorders[39], but its levels are rarely in accordance with clinical performance in enteritis[41].

Considering the unstandardized range of commercial assays, fecal calprotectin is rarely a suitable choice for intestinal evaluation in China[42]. Additionally, the combined analysis of fecal calprotectin, serum matrix metalloproteinase 9, and interleukin-22, which are strongly associated with intestinal inflammation, are integrated markers in IBD[43]. We identified SPINK4 as a novel histological and serological biomarker of IBD that can distinguish disease from healthy

**Fig. 3 | EGFR pathway is the intermediary agent for SPINK4 participating in GC differentiation. a** Relative expression of IEC markers from different branches, including enteroendocrine cells (*Neurog3, Insm,* and *Pax4*), secretory lineage (*Muc2, Gfi1, Foxa2, Reg3a,* and *Lyz*), M cells (*Spib*), Tuft cells (*Dclk*), stem cells (*Olfm4, Lgr5*), and absorptive cells (*Si*) in mouse organoids with rSPINK4 stimulation. The degree of color depth indicates the abundance of expression positively. **b** Levels of phosphorylated and total EGFR in NCM460 and HT-29 cells with 5 ng/mL rSPINK4 stimulation for 0–60 min. **c** Direct downstream molecules of the EGFR pathway detected using western blotting with rSPINK4 stimulation for 0–60 min. **d** Typical fluorescent images of intestinal organoids from WT and KO mice including MUC2 (green), E-cadherin (red), and DAPI (blue), with rSPINK4 (100 ng/mL) and AG1478 (10 μM) stimulation under inflammatory conditions (50 ng/mL TNF-α treatment). **e** Quantitative analysis of weight loss with SPINK4 and gefitinib

intervention following DSS administration in the control group drinking water (control, $n = 4$); DSS and PBS administration group (DSS + PBS, $n = 6$); DSS and rSPINK4 administration group (DSS + rSPINK4, $n = 5$); and DSS, rSPINK4, and gefitinib administration group (DSS + rSPINK4 + gefitinib, $n = 4$). **f** Representative canonical endoscopic images with the lesion circled with white arrows. **g** Quantification of colonic length. **h** The levels of the phosphorylated and total EGFR were determined in WT and KO mouse tissues (upper panel) and rSPINK4 treatment group (lower panel). Data are presented as the mean ± SEM. All tests were two-sided. Statistical significance was calculated using unpaired Student's $t$ test (**h**). Besides, Kruskal–Wallis test (**h**) and one-way ANOVA (**b**, **e**, **g**) were performed for multiple comparison; $n = 3$–5 biologically independent experiments (**b**, **g**, **h**). Source data are provided as a Source Data file.

conditions. The expression pattern of SPINK4 in acute colitis models induced by DSS differed from those observed in other models and IBD patients previously mentioned. The significant decrease in SPINK4 in active stage suggests its potential occurrence during the specific clinical process of acute colitis or the preclinical phase of IBD, particularly UC, which could be attributed to the dramatical depletion of goblet cells during the onset of colitis. Additionally, the SPINK4 level in sera could be used to assess the GCs status and partially substitute endoscopy with high specificity. Due to the hysteresis properties of endoscopic performance, patients may show pathological changes and elevation of SPINK4 levels without an observable shift in appearance, which was consistent with the relatively low sensitivity of SPINK4 for endoscopic activity evaluation; therefore, we pursued an objective of "histological healing" in IBD[44], defined as disease remission under endoscopy without histological inflammation. Further evidence needs to be produced about SPINK4 expression in other colitis pathologies, such as Behcet's disease.

Along with more discovery in IBD, targeting molecules are enrolled in clinical trials. Antibodies against TNF-α are widely employed in refractory CD patients; nevertheless, some patients do not respond to this targeted therapy[45]. Herein, we proposed a new treatment using rSPINK4 that is predicted to have few side effects in colitis. Owing to the abundance of SPINK4 in the intestine, it is a promising candidate for targeted therapy for colitis. The EGFR pathway, as a classical cascade, participates in epithelium regeneration and carcinoma formation, but its function in IBD development remained undiscovered[46]. Endogenous EGF is an essential ligand of EGFR, whose production has no alteration in IBD tissues[47]. SPINK4 interacted with EGFR and activated a cascade reaction for intestinal repair under inflammation attack rather than other ligands. Considering a synergistic effect with infliximab treatment, rSPINK4 may show clinical potential in both monotherapy and combination treatment.

Although, Pam2CSK4, as a key element of innate immune response, was affirmed to trigger SPINK4 production, it remains unclear which microbial components contribute to this process. Additionally, the subset producing SPINK4 was identified as a population at the initial stage, which was classified as a GC population according to the scRNA-seq analysis. The characteristics and function of this group remain to be elucidated. Our results suggest SPINK4 as a receptor agonist of EGFR targeting the epithelium recovery, and participating in the colitis pathogenesis. The regulatory role of SPINK4 in intestinal epithelial regeneration warrants further investigation in *Egfr*-cKO mice. The mechanism of SPINK4 focused on the mucus property remains to be verified.

In conclusion, we identified SPINK4 as not only a promising biomarker for the identification of IBD and estimation of the specific endoscopic status in colitis but also a target to ameliorate colitis by rebuilding GCs and the mucus layer via EGFR signaling and its downstream effectors. The identification of SPINK4 is a potential breakthrough in the diagnosis and treatment of IBD pathologies.

## Methods

### Specimen collection
Human serum and colonic samples were collected from healthy controls and IBD patients through the clinical process conducted at the First Affiliated Hospital of Sun Yat-sen University. The diagnosis and Montreal classification of IBD was based on the European Crohn's and Colitis Organization Guidelines and verified after multi-disciplinary treatment from diversified divisions.

### Single-cell RNA-sequencing (scRNA-seq)
The collection of biopsies and sample analysis are mentioned in detail in a previous study[48]. Five UC patients and four age- and sex-matched healthy controls were enrolled. The inflamed region at the sigmoid and uninflamed region at the ascending colon from the same patient respectively served as UC and self-control (SC) groups. Biopsy specimens collected from the sigmoid region of healthy controls served as the HC group. A Chromium Single-Cell Controller (10x Genomics, California, USA) and an S1000 Touch Thermal Cycler (Bio-Rad, California, USA) were operated for single-cell-bead generation and reverse transcription. The scRNA-seq libraries were prepared using Illumina NovaSeq 6000 sequencer (Illumina, San Diego, CA). After the reads processing and quality control, the cluster cells were analyzed by Seurat v3.1 integration workflow with SCTransform normalization method and visualized using the Uniform Manifold Approximation and Projection (UMAP) plot. R package monocle3 v0.2.1 and velocyto.R v0.6 were used for conducting the pseudotime analysis.

### RNA sequencing and analysis
Human intestinal mucosa samples were collected from healthy volunteers ($n = 18$), UC patients ($n = 9$), and CD patients ($n = 11$) and screened according to the inclusion and exclusion criteria for transcriptome sequencing (OE Biotech, Shanghai, China). Total RNA was extracted from the human endoscopic samples using TRIzol (Invitrogen, CA, USA) according to the manufacturer's protocol. We used 1 μg RNA to generate cDNA libraries with a TruSeq RNA Library Prep Kit (Illumina, USA) and then sequenced them on a HiSeq instrument (Illumina). The differential mRNA expression was analyzed and presented as a heatmap using Origin 2022 software (Supplementary Fig. 1a). A two-tailed Student's $t$ test and one-way analysis of variance (ANOVA) were used to analyze these data.

### Cell culture system
HT-29 (human colorectal epithelial adenocarcinoma cell line), Caco-2 (human colorectal epithelial adenocarcinoma cell line) and LS174T (human colonic epithelial adenocarcinoma cell line) cells were purchased from the American Tissue Culture Collection (ATCC, VA, USA) and cultured in ATCC-formulated McCoy's 5a Medium Modified (ATCC) and ATCC-formulated Eagle's Minimum Essential Medium (ATCC) supplemented with 10% fetal bovine serum and 1× penicillin–streptomycin (Invitrogen) in a 5% $CO_2$ atmosphere at 37 °C. The NCM460 cells, a human normal-derived colonic mucosal

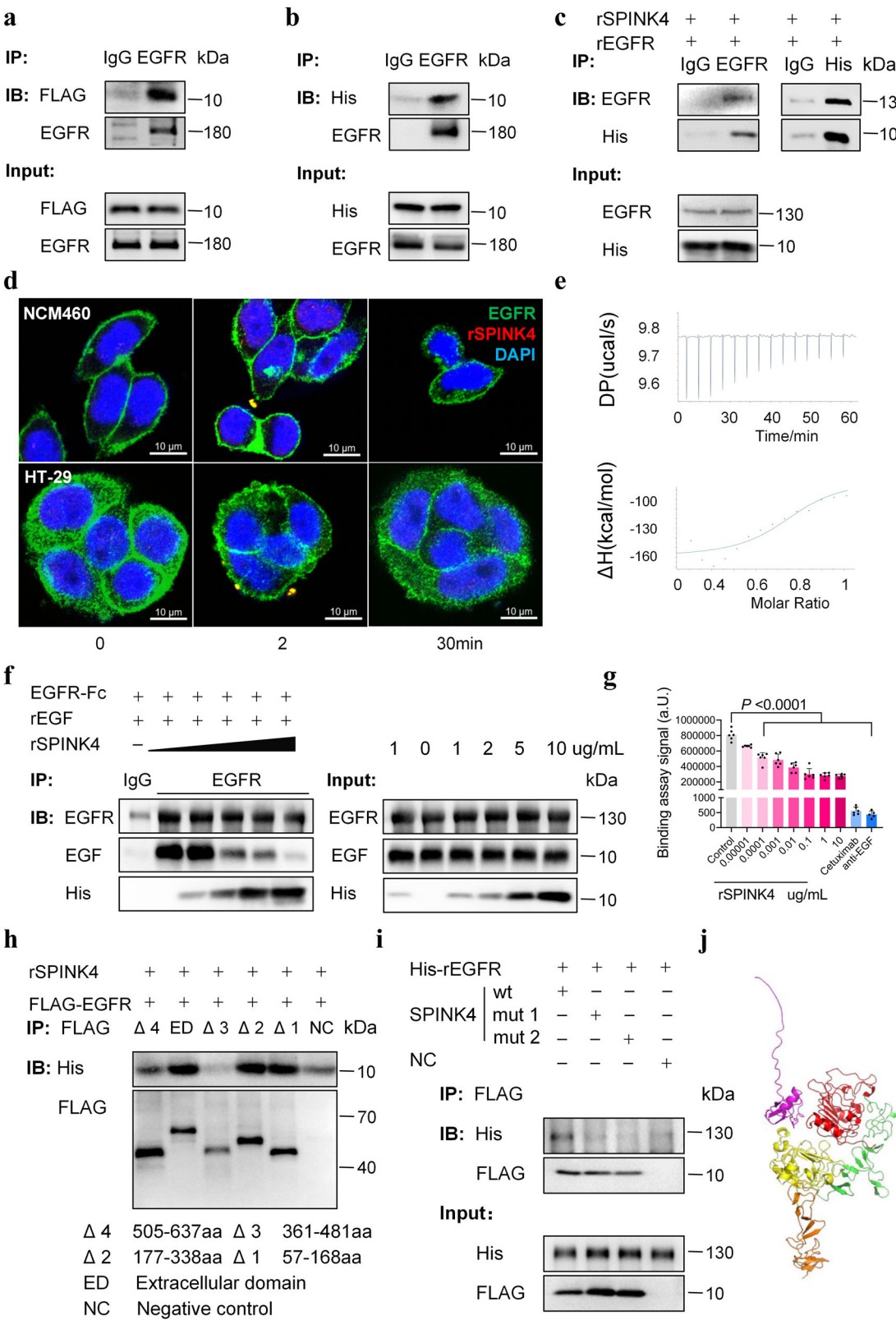

epithelial cell line (INCELL, TX, USA) were incubated in M3: BaseF medium (INCELL) containing 10% fetal bovine serum and 1× penicillin–streptomycin. LS174T cells were co-cultured with cytokines and bacterial components at different times at varied gradient concentrations. Human-derived recombinant SPINK4 protein (rSPINK4) was added to cultures at different concentrations (0, 5, 20, and 100 ng/mL) with AG1478 (10 μM, MCE, USA) to stimulate NCM460 and HT-29 cells from 2 min to 24 h.

**Lentivirus infection and transfection**

Lentivirus and negative control were designed and constructed by OBiO Technology (Shanghai, China). According to the instruction of the manufacturer's lentivirus operation manual, the transfection of LS174T cells was accompanied by the addition of 10 μg/mL polybrene after cells were seeded at the density of $0.5 \times 10^5$ cells/cm2. The culture medium was refreshed without lentivirus or polybrene after 24 h, and cell-stable clones for further culture were maintained in a medium

**Fig. 4 | SPINK4 can mediate its function in GC differentiation by interacting with EGFR. a** The FLAG-SPINK4 protein, which exists in the concentrated medium of SPINK4- overexpressing cells, is involved in the EGFR protein from cell lysis combined with EGFR antibody. IgG is the contrast to EGFR antibody. **b** The binding test of rSPINK4 protein and EGFR in membrane extraction under the incubation in vitro. **c** rSPINK4 coprecipitates with rEGFR (extracellular part) mutually in the in vitro pull-down assay controlled by IgG. **d** Representative immunofluorescent image of rSPINK4 and EGFR localization (EGFR: green, rSPINK4: red, DAPI: blue). **e** The curve of released heat and change in △H are performed when the rEGFR-His protein (6 μM) is titrated by rSPINK4-His protein (30 μM) via the ITC assay. **f** The pull-down assay of rEGF and rEGFR (extracellular part) in a SPINK4-dose-dependent manner. The concentration of rSPINK4 is shown. **g** A typical binding curve between biotinylated EGF and EGFR was performed using AlphaLISA binding assay. **h** Image profile of the interaction between rSPINK4 and the deleted extracellular domain of EGFR (△1: the deletion of 57–168 aa, △2: the deletion of 177–338 aa, △3: the deletion of 361–481 aa, △4: the deletion of 505–637 aa, ED: whole extracellular EGFR, NC: vector). **i** Interaction between mutant rSPINK4 (wt: whole sequence of SPINK4, mut1: mutation sites: C65A, C68A, and C86A, mut2: mutation sites: Q48A and M49A, NC: vector) and extracellular EGFR in vitro. **j** Predictive interactive model of SPINK4 and extracellular EGFR (pink: SPINK4, the tail is at N-terminal; red: EGFR subdomain including 57–168 aa; green: EGFR subdomain including 177–338 aa; yellow: EGFR subdomain including 361–481 aa; orange: EGFR subdomain including 505–637 aa). Data are presented as the mean ± SEM. The Statistical test was two-sided using one-way ANOVA (**g**); $n = 6$ biologically independent experiments (**g**). Source data are provided as a Source Data file.

containing 4 μg/mL puromycin. Transient transfection was achieved via single-guide RNA (sgRNA) of *SPINK4* and *EGFR* synthesized by OBiO Technology. LS174T cells were transfected with sgRNA using Lipofectamine 3000 (Invitrogen) following the manufacturer's protocol. The monoclonal *SPINK4* or *EGFR* knocked-out cell lines were acquired through single-cell proliferation.

The sequences of sgRNAs targeting *SPINK4* and *EGFR* were as follows:

*SPINK4* 5′-CTTGGCTGCCCTCCTTGTTGTGG-3′
*SPINK4* 5′-CGCAGACCAGGTTGGACATCTGG-3′
*SPINK4* 5′-ACCTGGTCTGCGGCACTGATGGG-3′
*SPINK4* 5′-CCGCCAGTGGGTAATCGCCCTGG-3′
*EGFR* 5′-CTTCGCACTTCTTACACTTG-3′
*EGFR* 5′-CTCGTGCGTCCGAGCCTGTG-3′

### Cell counting kit-8 assay

The cell counting kit-8 (CCK-8, Keygene Biotech, China) test was conducted following the protocol provided by the manufacturer. Cells were seeded at the density of $1-5 \times 10^5$/mL in 96-well plates and harvested for 12–24 h. We then added 0, 5, 20, and 50 ng/mL rSPINK4 to the medium for 24 and 48 h with or without 50 ng/mL tumor necrosis factor alpha (TNF-α). After incubation for 4 h, the absorbance of the reactive solution was measured at 450 nm using microplate reader (Tecan, Switzerland).

### Immunohistochemistry, immunofluorescence, and immunocytochemistry

Tissue sections for immunohistochemistry and immunofluorescence analysis were acquired from paraffin-embedded tissues. After being deparaffinized and gradient-hydrated, the sections were processed for antigen recovery and incubated with 3% hydrogen peroxide for 15 min to expose the epitope and quench the endogenous peroxidase activity. The sections were blocked with 3% bovine serum albumin (BSA) in phosphate-buffered saline (PBS) for 1 h and incubated with anti-SPINK4 for humans (1:250, Invitrogen), anti-SPINK4 for mice (1:500, personally produced by Genetex, USA), anti-MUC2 (1:100, Genetex), anti-EGFR (phosphor-Y1068, 1:500, Abcam), anti-active YAP1 (1:2000, Abcam) and anti-β-catenin (1:2000, Proteintech, USA) at 4 °C overnight. After being rinsed with PBS−0.1% Tween-20 three times, the sections were incubated with horseradish peroxidase (HRP)-conjugated secondary antibodies (Cell Signaling Technology, USA) for 30 min at room temperature (RT). Counterstaining with hematoxylin, dehydrating in serial ethanol dilutions, and mounting with coverslips were performed before imaging. For immunofluorescence assays, the sections were incubated with Alexa Fluor anti-rabbit, anti-rat IgG (H + L, 1:400, Invitrogen) at RT for 1 h and then mounted in antifade mounting solution with 4′,6-diamidino-2-phenylindole (DAPI, Invitrogen). Images were taken with an Olympus BX-63 microscope (Japan).

Cells for immunocytochemistry were processed similarly to those in the immunofluorescence assay by first fixing them in 4% paraformaldehyde for 15 min and then permeabilizing with 0.5% Triton-X-

100 for 15 min. The samples were blocked and incubated in anti-EGFR (1:1500, Proteintech). After incubation with Alexa Fluro anti-mouse, anti-rabbit (1:400, Invitrogen), the samples were counterstained with DAPI and imaged with a Zeiss LSM780 confocal microscope.

### Quantitative reverse transcription-polymerase chain reaction (qRT-PCR)

Total RNA was extracted from cells or biopsies using TRIzol Reagent (Invitrogen) according to the manufacturer's protocol. The RNA concentration was measured with Nanodrop 2000 (Invitrogen). The Transcriptor First Stand cDNA Synthesis Kit (Roche, Switzerland) was used for the reverse transcription of cDNA in vitro. qRT-PCR test was performed using Fast Start Universal SYBR Green Master Mix (Roche, Switzerland). The expression of target genes was measured relative to *GAPDH* or *β-actin* based on the delta-delta Ct method.

### Western blotting

Cell or tissue extractions were performed in RIPA buffer (CST, USA) supplemented with protease and phosphatase inhibitor cocktail (Invitrogen) and SDS-polyacrylamide gel electrophoresis was performed as previously described[49]. After electrophoresis and translocation of the proteins, the membrane was incubated with skim milk for 1 h and antibodies at 4 °C overnight, followed by secondary antibody incubation for 1 h. The antibodies used are listed in supplementary information file (Supplementary Table 2). The HRP substrate (Millipore, Germany) was used to visualize the immunoblotting. Chemiluminescence signals were acquired with iBright FL1500 imaging system (Invitrogen). GAPDH or β-actin image was set for normalization. Semiquantitative assessment of band density was conducted by Image-Pro plus and visualized by bar chart.

### Base Scope assay

Base Scope® probes targeting *SPINK4* were designed and synthesized by Advanced Cell Diagnostics (ACD, USA) and used with Base Scope Assay Kits (ACD). The fresh tissue sections for no more than 2 weeks were deparaffinized, and gradient-hydrated according to the manufacturer's protocol. The sections were treated with hydrogen peroxide and retrieval solution, followed by pretreatment with protease before being hybridized with probes. An amplifying system using a specific cascade process was employed to detect RNA (Base Scope Duplex AMP 1-12). After the signal was amplified, the sections were counterstained with hematoxylin and mounted with Vector Labs Vectamount (ACD, USA). The Olympus BX-63 microscope was used for image collection.

### Fluorescence in situ hybridization

The universal bacterial probe EUB338 (5′-GCTGCCTCCCGTAGGAGT-3′) for fluorescence in situ hybridization (FISH) test was designed and synthesized by IGE Biotechnology Ltd (Guangzhou, Guangdong, China). The assay was performed as previously reported[6,50]. The FISH hybridization solution included 20 mM Tris-HCl (pH 7.4), 0.9 M NaCl,

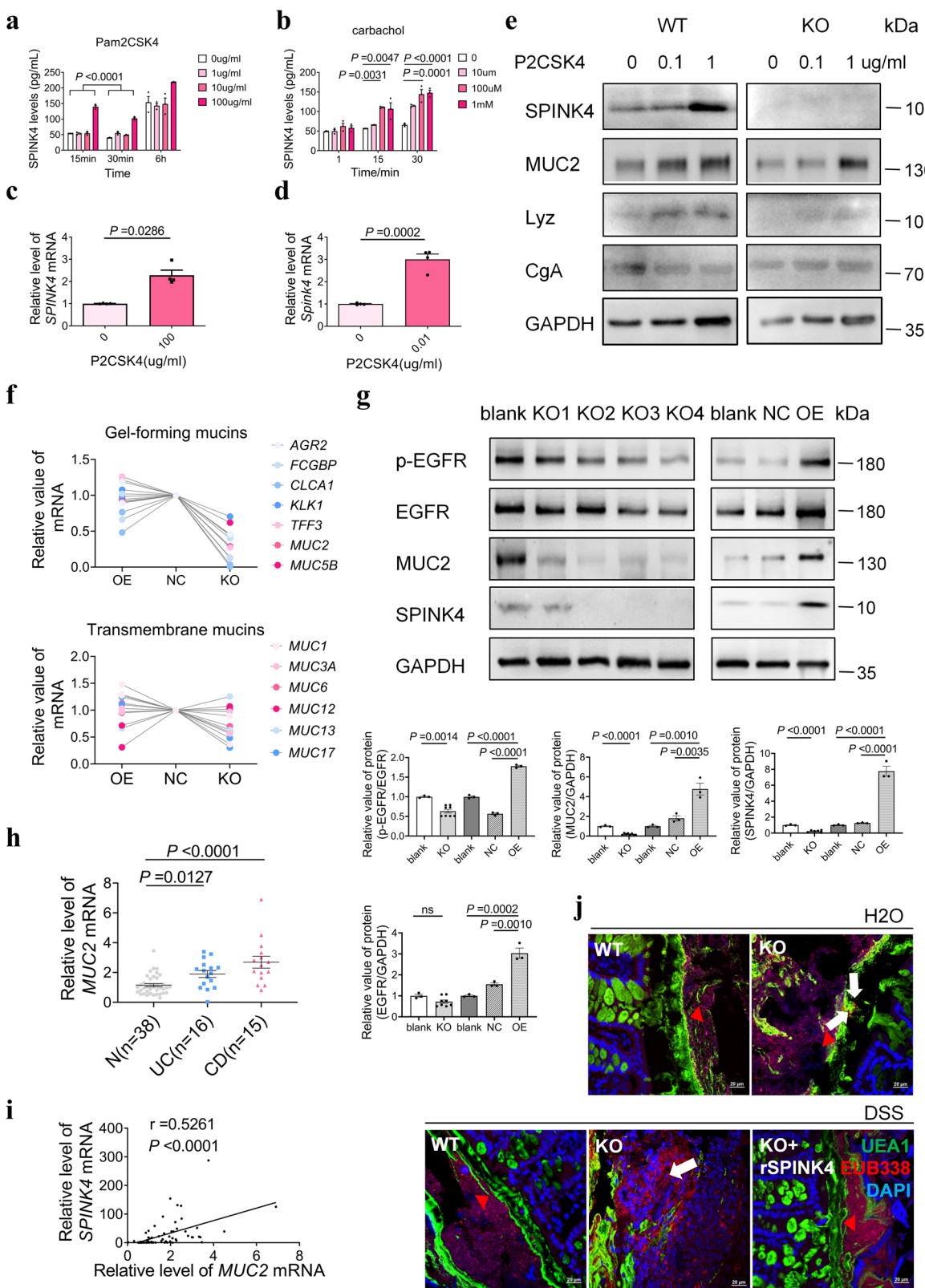

and 0.1% sodium dodecyl sulfate (SDS, w/v) in nuclease-free water with 25% formamide (v/v). The FISH washing buffer (20 mM Tris-HCl [pH 7.4] and 0.9 M NaCl) was prepared ahead. The deparaffinized sections were surrounded by a hydrophobic circle made with a PAP pen (Vector, USA) and then hybridized with pre-warmed probe solutions at 50 °C overnight. After rinsing with the washing buffer, the sections were counterstained with Hoechst 33342 (1:1000 in PBS, Invitrogen) and

40 μg/mL mucus stain UEA-1 (Vector) at 4 °C for 45 min in the dark. The stain was visualized with a Zeiss LSM780 confocal microscope.

**Co-immunoprecipitation and pull-down assays**
Co-immunoprecipitation (Co-IP) and pull-down assays between SPINK4 and EGFR were conducted as previously described[22]. The medium from SPINK4-FLAG-overexpressing NCM460 or Caco-2 cell cultures was

**Fig. 5 | Microbiota induce the production and secretion of SPINK4 to enhance mucus abilities in colitis. a**, **b** SPINK4 levels in LS174T cell supernatant stimulated with Pam2CSK4 (**a**) and carbachol (**b**). **c**, **d** Relative expression of *SPINK4* with 100 μg/mL Pam2CSK4 stimulation in human intestinal organoids (**c**) and with stimulation with a lower concentration (0.01 μg/mL) in mouse intestinal organoids (**d**). **e** SPINK4 expression presented with MUC2, Lyz, and CgA values in the intestinal organoids from WT and cKO mice with Pam2CSK4 treatment. **f** Gel-forming mucin and transmembrane mucin expression when *SPINK4* was overexpressed or knocked out were assessed using qRT-PCR test. OE, overexpression; NC, negative control; KO, knockout. **g** MUC2 expression was influenced by the intrinsic SPINK4 levels in the LS174T cells and the phosphorylated EGFR levels. OE, overexpression; NC, negative control; KO, knockout. Blank indicates no vector involved. **h**, **i** The

correlation between *MUC2* level with *SPINK4* expression in IBD. The mRNA level was normalized to β-actin level (**h**). The Pearson correlation coefficient (*r*) was used to estimate the correlation between *MUC2* and *SPINK4* expression (**i**). **j** The apparent difference in colonic permeability following *Spink4* knockout (UEA1: green, EUB338: red, DAPI: blue; white arrows: the bacteria which penetrate the inner mucus layer; red triangles: the outside of the inner mucus layer. Data are presented as the mean ± SEM. All tests were two-sided. Statistical significance was calculated using unpaired Student's *t* test (**d**, **g**) and one-way analysis of variance (ANOVA) (**a**, **b**, **g**). Mann–Whitney *U* test and Kruskal–Wallis test were performed on non-normal data (**c**, **h**); *n* = 3 biologically independent experiments (**a**–**d**, **g**). Source data are provided as a Source Data file.

concentrated with ultrafiltration device (Millipore, Germany) at 16,000 × *g* for 30 min. Membrane proteins were extracted using a membrane protein extraction kit (Invent, USA) according to the manufacturer's instructions. The extract was incubated with the concentrated medium at 4 °C overnight on a rotator. Protease inhibitors, including aprotinin, bestatin, E-64, leupeptin, sodium fluoride, sodium orthovanadate, sodium pyrophosphate, and β-glycerophosphate (Invitrogen) were added to stabilize the mixture. A combination of anti-EGFR magnetic beads (Invitrogen and Proteintech) was added to the mixture and incubated at 4 °C overnight, followed by washing in TBS-T (TBS solution with 0.1% Tween-20). Next, the bead–protein mixture was eluted in 2× loading buffer at 95 °C. Human-derived rSPINK4 was substituted for the medium with the SPINK4-FLAG protein and precipitated using the anti-EGFR antibody-binding beads.

In the pull-down assays, 200 ng/mL human-derived rSPINK4-His (Sino Biological) and 1000 ng/mL Fc-tagged extracellular section of EGFR (rEGFR, Sino Biological) were incubated in 1 mL PBS with protease inhibitors at 4 °C overnight. The protein mixture was then incubated with anti-His (MCE) or anti-EGFR magnetic beads. The samples were washed and examined by Western blot analysis. In the competitive immunoblot assay, 1000 ng/mL human-derived-rEGFR with an Fc tag was incubated with 1000 ng/mL EGF (Peprotech, USA) in a rSPINK4-His (Sino Biological)-dependent manner (0, 1, 2, 5, 10 μg/mL). Anti-EGFR or IgG magnetic beads (MCE) were enrolled in further immunoprecipitation test.

The production of the transcription and translation system in vitro was also used with recombinant protein in vitro, as mentioned before. Unlike the procedure of pull-down assay, we selected anti-FLAG magnetic beads rather than anti-His.

## Colocalization analysis of rSPINK4 and EGFR

A total of 10 μg human-derived rSPINK4-His (MCE) was labeled with a fluorescent dye using the Alexa Fluor® 555 Microscale Protein Labeling Kit (Invitrogen) following the manufacturer's protocol. Alexa Fluor 555 dye-labelled SPINK4 complexes were measured at 0.88 mg/mL using a Nanodrop 2000. NCM460 and HT-29 cells were stimulated with the SPINK4-conjugate for 10–30 min and then cultured for another 5 min without stimulation. The colocalization analysis was performed using a confocal LSM780 microscope after the immunocytochemistry assay.

## ITC assays

The rSPINK4-His and rEGFR-His (extracellular domain) proteins were enrolled, and the MerryBio Co. Ltd. MicroCal PEAQ-ITC machine (Malvern MicroCal, UK) was used to conduct this assay.

We harvested the protein supernatant collected by centrifugation at 1000 × *g* for 10 min to remove insoluble components and thoroughly cleaned the ITC machine with diluted Decon90, ultrapure water, and chromatographic-grade methanol. The rSPINK4-His protein (30 μM in 100 μL) was set as a titrating solution and loaded into the titration needle, whereas rEGFR-His protein (6 μM in 300 μL) was added to the sample pool. During the titration process, the heat was released and recorded by the machine detector, which was accurately controlled by the titration between 30 μM rSPINK4-His with PBS

solution. The KD value was defined as a binding constant, which measures the strength of a reversible reaction. △H, as enthalpy change, was calculated automatically with the KD value using the MicroCal PEAQ-ITC Analysis Software.

## AlphaLISA binding assay

The quantitative assessment of EGFR ligand competition was conducted using AlphaLISA EGF and EGFR (Human) Binding Kit (PerkinElmer, USA) in accordance with the manufacturer's manuals. Anti-EGF antibody (Abcam) and Cetuximab (MCE) were also employed. The serial dilutions of antibody or sample were mixed with anti-Human IgG Fc-specific AlphaLISA acceptor beads (20 μg/mL) and human-derived rEGFR (3.125 nM). After consecutive incubation at 23 °C for 30 min, biotinylated EGF (3.125 nM) and streptavidin donor beads (20 μg/mL) were sequentially added. The signal was detected by Varioskan LUX (Invitrogen) equipped with the AlphaLISA option using the following settings: total measurement time: 550 ms, laser: 680 nm, excitation time: 180 ms, delay time 20 ms, emission filter: wavelength 570 nm.

## Transcription and translation in vitro

The TNT® T7 Quick Coupled Transcription/Translation System (Promega, USA) was used to induce the transcription and translation of plasmids or PCR products in vitro. The design and construction of plasmids for transcription was based on the pcDNA 3.1(+)−3×FLAG vector. We constructed the plasmids for the EGFR extracellular domain with different deleted subdomains and SPINK4 with missense mutations (C65A, C68A, and C86A; Q48A and M49A). A mixture of plasmids, Master Mix, and methionine was incubated for 90 min at 30 °C in a water bath. The amount of protein production was ~3–6 μg/mL. The production of the EGFR extracellular domain was supplemented with canine pancreatic microsomal membranes (Promega) that were suitable for the membrane proteins during post-translation processing in vitro.

## Isolation of crypts and organoid culture

Organoids were generated as previously reported[51]. Basic mouse intestinal organoid medium was composed of Advanced DMEM/F12 (Invitrogen), 2 mM GlutaMax (Invitrogen), 10 mM HEPES (Invitrogen), and 1× penicillin/streptomycin (Invitrogen) and supplemented with 1× N2 (Invitrogen), 1× B27 (Invitrogen), 1 mM *N*-acetyl-L-cysteine (Invitrogen), and 20% R-spondin1-conditioned medium. In addition to these components, 50 ng/mL mouse EGF (Peprotech, USA), and 100 ng/mL mouse Noggin (Peprotech) were also added for the culture of mouse intestinal organoids. The human organoids were also supplemented with 50% Wnt3a-conditioned medium, 10 mM nicotinamide (Sigma-Aldrich, USA), 10 μM SB202190 (Selleck, USA), 0.5 μM A8301 (Sigma-Aldrich), and 10 nM [Leu15]-gastrin I (Sigma-Aldrich)[52]. The Wnt3a- and R-spondin1-conditioned media were collected as previously described[53]. Owing to the essential roles of EGF, Noggin, and R-spondin1 in organoid development, we named the complete medium as ENR. After being washed thoroughly with Dulbecco's PBS without Ca²⁺ or Mg²⁺ thoroughly, the intestinal samples harvested from human or mice were incubated in 2 mM ethylenediaminetetraacetic acid buffer for 30 min at

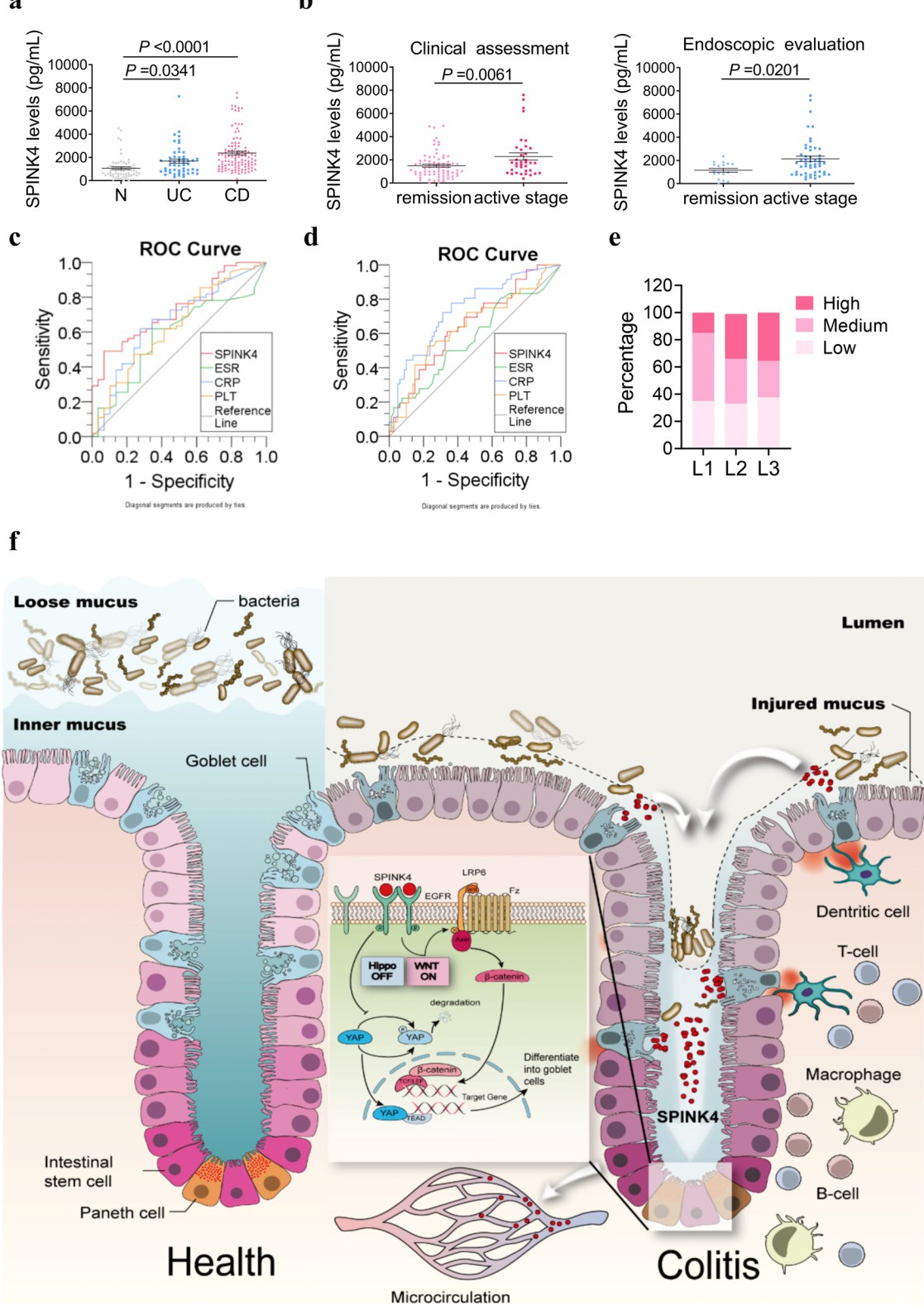

**Fig. 6 | Circulating SPINK4 from GCs plays an important role in the screening of colitis disease activity for IBD patients. a** Circulating SPINK4 level from healthy controls ($n = 64$), UC ($n = 51$), and CD patients ($n = 108$) detected using ELISA. **b** SPINK4 levels between remission and active stages, based on clinical characteristics (left) or endoscopic performance (right) in the sera of IBD patients. **c, d** The efficiency of SPINK4 (red line), ESR (green line), CRP (blue line), and PLT (orange line) values for endoscopic (**c**) or clinical (**d**) activity assessment are shown via ROC curve. **e** Percentage of different level of SPINK4 in multiple intestinal locations including L1 (ileal), L2 (colonic), and L3 (ileocolonic), according to the Montreal classification. **f** Graphic abstract showing the involvement of extracellular SPINK4 in the regeneration of GCs via directly targeting EGFR pathway. Data are presented as the mean ± SEM. All tests were two-sided. Statistical significance was calculated using unpaired Student's *t* test (**b**, **c**) and one-way analysis of variance (ANOVA) (**a**). Source data are provided as a Source Data file.

**Table 1 | Basic clinical characteristics of the IBD patients and healthy controls**

| | N | UC | CD with classical therapy | CD with anti-TNF-α therapy |
|---|---|---|---|---|
| N | 64 | 51 | 43 | 65 |
| Male, n (%) | 33 (51.6) | 28 (54.9) | 22 (51.2) | 41 (63.1) |
| Median age, yr, (IQR) | 25 (24–28) | 42 (35–56) | 28 (25–36) | 25 (19.75–31.25) |
| BMI,kg/m², (IQR) | - | 19.6 (18.4–23.8) | 18.7 (16.8–20.5) | 19.4 (17.7–21.0) |
| Disease phenotype, n (%) | | | | |
| L1 | - | - | 11 (25.6) | 7 (10.8) |
| L2 | - | - | 0 (0) | 3 (4.6) |
| L3 | - | - | 30 (69.8) | 55 (84.6) |
| L4 | - | - | 2 (4.7) | 0 (0) |
| B1 | - | - | 19 (44.2) | 34 (52.3) |
| B1p | - | - | 7 (16.3) | 16 (24.6) |
| B2 | - | - | 7 (16.3) | 24 (36.9) |
| B2p | - | - | 2 (4.7) | 7 (10.8) |
| B3 | - | - | 17 (39.5) | 7 (10.8) |
| B3p | - | - | 8 (18.6) | 5 (7.7) |
| p | - | - | 17 (39.5) | 28 (43.1) |
| Disease extent, n (%) | | | | |
| Proctosigmoiditis | | 5 (9.8) | - | - |
| Left-sided colitis | | 14 (27.5) | - | - |
| Pancolitis | | 32 (62.7) | - | - |

4 °C with gentle shaking. Physical separation was used to acquire crypts, followed by centrifugal sedimentation at 100–200 × *g* three times. The cell pellets were resuspended in Matrigel (Corning Life Sciences, Acton, MA, USA) with half-complete medium in 24-well plates and cultured in the ENR medium. The medium was changed every 2–3d and passaged after 7–10 d. The organoids were used after 2–5 passages for the experiments. The organoids generated from both human and mouse tissues were cultured in the medium without EGF, which synergistically affects the EGFR pathway, and supplemented with 100 ng/mL rSPINK4 and 10 µM AG1478 (MCE, USA) for 48 h. The inflammatory condition could be developed using 50 ng/mL TNF-α. Cells were then imaged in a bright field using a Leica DMi8 microscope (Germany).

## Whole-mount staining

Whole organoids were harvested in cell recovery solution (Corning, USA)[54], followed by fixation in 10% formalin for 15 min, and permeated in 0.5% Triton-X for 20 min. After blocking in 1% BSA prepared in PBS, the organoids were incubated with antibodies against MUC2 (1:100, Genetex, USA), E-cadherin (1:100, Santa Cruz, USA), chromogranin A (CgA, 1:100, Santa Cruz), lysozyme (Lyz, 1:100, Santa Cruz) and Ki67 (1:250, Abcam, UK). After being washed in 1% BSA-PBS, organoids were labeled with Alexa Fluor anti-mouse, anti-rabbit or anti-rat IgG (H + L) (1:400, Invitrogen) at 4 °C overnight. Images were acquired using a Zeiss LSM780 confocal microscope after counterstaining with Hoechst 33342 (1:1000, Invitrogen).

## Enzyme-linked immunosorbent assay (ELISA)

After centrifuging at speed of 1900 × *g* and 16,000 × *g* for 10 min each, the sera from patients and healthy controls were stored at −80 °C. The cell supernatant was also collected after low-speed centrifugation. The concentration of SPINK4 was detected with an enzyme-linked immunosorbent assay (ELISA) kit (Sino Biologica, China) following the manufacturer's protocol. The absorbance of the reaction product was measured at 450 nm using a microplate reader. A SPINK4 level in sera

exceeding 1800 ng/mL was considered a high expression, whereas low expression was defined as a SPINK4 level less than 1000 ng/mL. The cut-off value was based on the tertiles.

## Mouse strains

The DSS- and TNBS-induced colitis mice were obtained from the BALB/c and C57BL/6 J strains, respectively, which were purchased from Gem-Pharmatech Co., Ltd. Villin-Cre mice were obtained from the Jackson Laboratory (USA) and the *Spink4*-floxed mice on C57BL/6 J background were generated by GemPharmatech Co., Ltd, based on the designed sgRNAs. The genotyping of mice was identified by DNA amplification before model construction. All in vivo mouse experiments were approved by the Ethics Committee of the First Affiliated Hospital, Sun Yat-sen University ([2020] No. 283). The feeding, operation and euthanasia protocols were supervised by the Animal Laboratory Center of the hospital, which complied with the principle of "3 R" for animal experiments.

The paired sgRNA sequences were as follows:
*Spink4*–5S3 TCGAATGACATGATCCGATA PAM: AGG
*Spink4*–5S4 AAGTCCTAGCTCTGCCTTAT PAM: CGG
*Spink4*–3S3 AGAGCCCGGATCAGCCACTG PAM: AGG
*Spink4*–3S4 CAAAGCCTCAGTGGCTGATC PAM: CGG

## DSS treatment and tissue assessment

Mice were treated with 3% dextran sulfate sodium (DSS, MP Bio-medicals, USA) with a molecular weight (MW) of 36,000–50,000 Da in drinking water for 7d according to the successful protocol (Fig. 2a)[55]. After the withdraw of DSS solution, the disease activity index (DAI) was measured as previously reported[55]. The endoscopic examination was carried out before animals were euthanized by cervical dislocation on days 9–11. The histological assessment used grades 0–3 of samples prepared with conventional paraffin embedding, sectioning, and hematoxylin and eosin (H&E) staining as previously described[55]. Additionally, Alcian blue and periodic acid Schiff (AB-PAS) staining was conducted as previously described[56]. Clinical and pathological activity assessment was evaluated by three independent investigators.

## TNBS treatment

The TNBS (Sigma-Aldrich)-induced colitis mouse model was generated as previously described (Fig. 2a)[55]. Following anesthetization with iso-flurane gas, 100 µL of 2.5% (w/v) TNBS in 50% absolute ethanol was administrated into the colon lumen 7d after pre-sensitization with 100 µL of 1% (w/v) TNBS. At the end of the experiments, mice were euthanized via cervical dislocation and samples were harvested in the active stage at 3–7d after enema and even after a few days for remission research. Histological assessment was conducted during this period.

## Application of rSPINK4-Fc, gefitinib, infliximab and Pam2CSK4 in the colitis model

After administration of DSS or TNBS, the mice were supplemented with 500 µg/kg mouse-derived rSPINK4-Fc or rFc (Sino Biological) by intra-peritoneal injection once per day from day 4 post DSS administration in DSS colitis models or 3 consecutive days after TNBS enema until the mice were euthanized (Fig. 2a). The tyrosine kinase inhibitor gefitinib (MCE) was used by gavage at 50 mg/kg in the last 3d in DSS-induced colitis model (Fig. 2a). Infliximab at 5 mg/kg by intraperitoneal injection for 2 consecutive days as treatment targeting TNF-α. Additionally, Pam2CSK4 (InvivoGen, France) was given by intraperitoneal injection at 250 µg/kg per day from the first day of DSS administration for 8 consecutive days. Assessment of disease activity was conducted as mentioned before.

## Flow cytometry targeting multiplex factors

The serum was harvested by gradient centrifugation, as mentioned before. The supernatant was stored at −80 °C. The inflammatory

factors were detected using a LEGENDplex kit (BioLegend, USA), and their levels were then measured using a flow cytometer.

## Myeloperoxidase activity test

Mouse tissues were obtained from the distal part of the colon and stored in liquid nitrogen for no >1 week. Myeloperoxidase (MPO) activity was used to estimate the activation level of neutrophils with an MPO assay kit (Nanjing Jiancheng Bioengineering Institute, China). The absorbance of the reaction product was measured at 460 nm using a microplate reader.

## In vivo intestinal permeability assessment

Fluorescein isothiocyanate-dextran (FD4, MW4000, Sigma-Aldrich, USA) was administrated by gavage at dosage of 22 mg/kg after the mice fasted for 4 h. The blood was collected by eyeball extirpating and then centrifuged at $1900 \times g$ for 10 min to collect the supernatant, which was centrifuged again at $16,000 \times g$ for 10 min. The final plasma and gradient-diluted standards were measured using a Spectra Max M5 with excitation/emission wavelengths of 490/530 nm (Molecular Devices, USA).

## Transmission electron microscopy

The intestinal samples from mouse models were cut into $1 \times 1 \times 1$ mm sections and immersed in ice-cold glutaraldehyde for no more than one month. The tissue was fixed with osmium tetroxide and gradient dehydrated with ethanol, and then embedded in epoxy resin. The sections were stained with 3% uranium acetate and lead citrate and visualized using a TEM (FEI Tecnai Spirit, USA).

## Endoscopic examination

We performed the mouse colonoscope using STORZ TC200 (KARL STORZ, Tuttlingen, German) with operating manual. After the mice received intraperitoneal injection of 50 mg/kg sodium pentobarbital, they were taped to a board tightly, especially at the tail, to clearly expose the anus. As the camera lens entered the intestine, the lumen could be observed with the middle of the colon being the maximal reach of the endoscope. Images were acquired throughout the duration of this procedure.

## Statistical analysis

The in vitro experiments were repeated at least three times, especially in representative experiments. The data were analyzed using Graph-Pad Prism and IBM SPSS Statistics 20 software, expressed as mean ± standard error of mean, and analyzed using an unpaired, two-tailed Student's $t$ test or one-way ANOVA for normal distribution. If not, non-parametric tests such as Mann–Whitney $U$ test and Kruskal–Wallis test were conducted. Correlations between *SPINK4* and *MUC2* levels were estimated using the Pearson correlation coefficient (r). The receiver operating characteristic (ROC) curve was obtained for the comparative analysis of sera. $p < 0.05$ indicated statistical significance.

## Study approval

The human and animal studies conducted in this study were approved by the Ethics Committee of the First Affiliated Hospital of Sun Yat-sen University ([2022] No. 317, [2020] No. 283). Informed consent was obtained from all participants. Procedures followed the Helsinki Declaration.

# Data availability

We confirmed that all data supporting our findings in this manuscript are available within the articles or Supplementary Information. The detailed original information could be acquired from the corresponding author upon request. The RNA-sequencing data were deposited into the NCBI Gene Expression Omnibus (GEO) database with the accession number GSE230113 [platform ID: GPL16791; dataset IDs: GSM7187739-GSM7187758]. Source data are provided with this paper.

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

## Acknowledgements

We thank the members of the multiple disciplinary teams and the Animal Laboratory Center at the First Affiliated Hospital, Sun Yat-sen University. We also appreciate all patients for participating in this study. This work was supported by grants from the National Natural Science Foundation of China (#82070538, #81870374, #81901447)

## Author contributions

S.Z. designed and oversaw the project, and revised the manuscript. Y.W. and J.H. performed the experiments and wrote the manuscript; G.Y., S.Z., and G.Z. analyzed the data and interpreted the results; X.C., X.L., G.L., and B.Z. conducted single-cell sequencing; Z.X., L.L., M.Z., X.L., and M.C. revised the manuscript. All authors read and approved the manuscript.

## Competing interests

The authors declare no competing interests.
