## [Peer Review File · Nature Communications]

REVIEWER COMMENTS

Reviewer #1 (Remarks to the Author):

The authors reported that SPINK4 is dynamically regulated in IBD patients and colitis model and specifically located in GCs. They also showed that SPINK4 serves as a serologic biomarker of IBD and has therapeutic potential for colitis. This is an interesting paper which newly suggests that TLR2/6 signaling-induced SPINK4 binds to EGFR and induces the differentiation of GCs, and that maintains intestinal homeostasis. However, there are several concerns which have to be addressed by the authors.

Major

(1) In the model mice, SPINK4 level in the active stage of colitis is lower than that in the remission stage. In human, SPINK4 level in the active stage of colitis is higher than that in the remission stage. How do the authors explain this discrepancy?

Minor

1) Although there are many genes whose expression was upregulated in single cell RNA sequencing, the reason why they focused on SPINK4 is not described. They should describe it.

2) Is the recombinant SPINK4 a human protein? Is it mouse protein? There is no description.

3) Did Conditional Spink4 KO mice have any phenotype?

4) The Murine Gene symbol of "SPINK4" should be "Spink4".

5) On page 7 line 150, "SPINK family members share a similar Kazal motif like EGF". The authors should show the reference.

Reviewer #2 (Remarks to the Author):

This experimental study deals with the role the secreted protease inhibitor SPINK4 may have in colitis, and especially in IBDs. The Authors described and proved its relationship with IBD in DSS- and TNBS-colitis mouse models and in humans, and linked their serum concentrations with

disease activity, with the highest levels of the molecule found in the remission stage of the disease. Moreover, they evaluated the therapeutic potential of the administration of recombinant SPINK4 in mice. It proved to be effective in reducing intestine inflammation per se and even having a synergistic effect when combined to anti-TNF α drugs. Most of the effect of SPINK4 is due to the activation of the EGFR pathway and the consequent induction of goblet cell differentiation. This enhances the mucus production, proved by the direct correlation with the increase of the MUC2 expression. Also, SPINK4 levels seem to be positively related to Pam2CSK4 ones, whose action on the other hand mimics the one of Gram negative bacteria LPS.

The amount of data brought by the Authors regarding this molecule made light on a new etiopathogenetical mechanism in IBDs, and potentially on a possible therapeutic application.

I ask the Authors to better clarify or explain the following points:

- 40-41: “The role of colonic GCs, typically characterized by mucin production, in colitis has been ignored”. Considering after this statement references regarding GC in IBD follow, especially number 15 and 19, I’d modify the “has been ignored” with “has been underestimated” or similar.
- 80-81: “however, SPINK4 was barely expressed in the active stage”. So was this difference statistically significant when compared to controls or not?
- 105-106: “This group had ... less reduced intestinal permeability”. I’d modify “less reduced” with a simpler “increased”, if so.
- 107-109: “we further tested GCs numbers and the mucus thickness and found that both of them in the rSPINK4 treatment group were obviously increased”. I’d suggest to remove the “obviously”
- 226-228: “Therefore, we cocultured LS174T, a colonic cancer cell line, which mimics GCs in vitro on account of abundant mucous vesicles, with microbiotic elements”. A quote is necessary.
- 228-229: “Pam2CSK4, a synthetic diacylated lipopeptide that activates the TLR2/TLR6229 heterodimer”. A quote is necessary.
- 231-232: “The other microbiota components”. The same reference I suggested for 226-228 should be quoted here too, to clarify which microbiota components the Authors are referring to.
- 245-246: “we estimated the mucus permeability, which is the most important intrinsic property of GCs”. Mucus permeability is not a property of GC, mucus production is.
- 245-255: “Moreover, there was a mild difference between the remission and active stages in CD patients”. I suggest to explicit in the text if this difference was statistically significant or not.
- 280-282: “The inability of IBD to be self-healing may be attributed to... other pathogenetic mechanisms”. A quote is necessary.
- 283-284: “Though numerous serological markers burst out”. A quote is necessary.
- 314: “which kind of microbiota”. I suggest to modify it in “which microbial composition” or “microbial components”.

- 334-337: “Five UC patients and four age- and sex-matched healthy controls were enrolled. The inflamed region at the sigmoid and uninflamed region at the ascending colon respectively served as UC and SC groups.” Why did the Authors decided to consider different regions for the two groups? Then, with “SC” do the Authors mean Healty Controls (HC)?; in case, the acronym must be changed.
- 337: “Biopsy specimens from healthy controls served as the HC group”: where the biopsies taken from the segments described in 336? I suggest to explicit it.

Reviewer #3 (Remarks to the Author):

This manuscript is focused on the regulatory effects of serine protease inhibitor Kazal-type 4 (SPINK4) on intestinal inflammation. The conclusions provided by the authors include: SPINK4 is associated with the colitis status in IBD patients; the increase in colitis induced by DSS and TNBS in mice with deletion of SPINK4 is rescued by recombinant SPINK4, which is synergistic with an TNF inhibitor, infliximab; SPINK4 promotes goblet cell differentiation through a Kazal-like motif-modulated EGFR-Wnt/ β -catenin and -Hippo pathways; microbiota-derived diacylated lipoprotein Pam2CSK4 triggers SPINK4 production. The author also stated that SPINK4 in circulation could serve as a standard to assess disease status in patients with IBD.

Although large amount of data is shown in this manuscript, useful data to support the conclusion is lacking.

1. More data from mice with EGF receptor specifically deleted in intestinal epithelial cells and EGF receptor knock out goblet cell lines are needed to support the involvement of EGF receptor signaling in SPINK4-regulated cellular responses.
2. The interaction of EGF receptor and SPINK4 identified by the co-immunoprecipitation and pull-down assays is not the strong evidence to support SPINK4 binding to the ligand binding site on EGF receptor to activate EGF receptor. EGF receptor ligand competition assay may support some evidence whether SPINK4 could function as a EGF receptor ligand. Thus, more studies are needed to elucidate the mechanisms by which SPINK4 activates EGF receptor.
3. There are no studies to investigate the effects of Pam2CSK4 on colitis and mucus production in vivo.
4. In the Method section, the authors stated “Villin-Cre mice were obtained from the Jackson Laboratory (USA) and the SPINK4 floxed mice were generated by GemPharmatech Co., Ltd, based on the designed sgRNAs”. No mouse background was provided. No experimental data to confirm the specific deletion of SPINK4 in intestinal epithelial cells in mice. No information about the phenotype of this transgenic mice. The authors defined these mice as “conditional knockout mice” in this manuscript. Is the gene mutation in floxed SPINK4-Villin-Cre mice conditional or constitutive?

5. In the result section, only conclusion from each figure is provided. The authors should present detailed data and explanation to support the conclusion.
6. No statement about how many times of the experiments are repeated for in vitro studies.
7. Higher magnification images of HE stained colonic sections are needed to examine the histological changes (Figure 2O).
8. It is difficult to get any useful information from endoscopic images (Figure 2K and Figure 3F).
9. It is not clear whether colonic epithelial cells or total colonic tissues are used for the Western blot analysis of Muc 2 levels (Figure 2J) and EGF receptor phosphorylation (Figure 3H). The molecular weight of Muc2 is identified as around 130kDa in this manuscript. The authors need to confirm it.
10. Fold change of band density in Western blot from all in vitro experiments and mice should be provided.

Response Letter to Reviewers:

We would like to thank the reviewers for their time and thoughtful comments on our paper. In the following, we respond to each of the concerns and recommendations. Comments are indicated in blue and our responses are presented in black. Please see below our point-by-point responses and corresponding changes in the manuscript, which are highlighted in yellow with red font.

Reviewer #1 (Remarks to the Author):

The authors reported that SPINK4 is dynamically regulated in IBD patients and colitis model and specifically located in GCs. They also showed that SPINK4 serves as a serologic biomarker of IBD and has therapeutic potential for colitis. This is an interesting paper which newly suggests that TLR2/6 signaling-induced SPINK4 binds to EGFR and induces the differentiation of GCs, and that maintains intestinal homeostasis. However, there are several concerns which have to be addressed by the authors.

Major

Comment 1: In the model mice, SPINK4 level in the active stage of colitis is lower than that in the remission stage. In human, SPINK4 level in the active stage of colitis is higher than that in the remission stage. How do the authors explain this discrepancy?

Response 1: Thank you for your constructive question. Inflammatory bowel disease (IBD) is a chronic and refractory inflammatory gastrointestinal disease characterized

by immune system dysfunction. Colitis mice administrated by TNBS and DSS have been generally recognized as a disease model for IBD¹. Given the intestinal feature, the chemically induced chronic colitis model is widely acknowledged as a reliable *in vivo* model for studying IBD^{2,3}. Furthermore, the acute model recapitulates the development of the mucosal injury in a simple and inexpensive setting and partially reflects the underlying process of IBD at active stage¹. In this study, we preferred to validate the expression of SPINK4 in colitic intestine based on four models administrated with TNBS and DSS. The expression pattern of SPINK4 in patients was also observed in the acute models induced by TNBS and the chronic colitic models, which could better represent the natural course of CD. The acute DSS-induced models, widely recognized as a suitable model for UC¹, exhibited a significant decrease in SPINK4 expression in the active stage. This could potentially occur during the specific clinical process of acute colitis or the preclinical phase of IBD, especially UC. This might be attributed to the drastic depletion of goblet cells during the onset of UC, which was in line with the pseudotime analysis of RNA-seq data based on endoscopic specimen from UC patients. Accompanied by the intestinal disorders in IBD, TLR2/6-induced SPINK4 expression is triggered to accelerate epithelium recovery, which can be identified as active IBD in the clinical stage. We have discussed this issue in the revised text. (p.15, para.2; line 321–326)

Reference

1. Wirtz, S., *et al.* Chemically induced mouse models of acute and chronic

- intestinal inflammation. *Nat. Protoc.* **12**, 1295-1309 (2017).
2. Pizarro, T.T., Arseneau, K.O., Bamias, G. & Cominelli, F. Mouse models for the study of Crohn's disease. *Trends Mol. Med.* **9**, 218-222 (2003).
 3. Holgersen, K., *et al.* High-resolution gene expression profiling using RNA sequencing in patients with inflammatory bowel disease and in mouse models of colitis. *Journal of Crohn's & colitis* **9**, 492-506 (2015).

Minor

Comment 2: Although there are many genes whose expression was upregulated in single cell RNA sequencing, the reason why they focused on SPINK4 is not described. They should describe it.

Response 2: Thank you for your suggestion. In our study, we attempted to figure out the critical factors in goblet cell regeneration involved in IBD. We identified SPINK4 as a goblet cell biomarker using scRNA-seq, as it is significantly overexpressed in colitic intestine. Combined with the analysis of RNA sequencing and GEO database ¹, SPINK4 performs as an excellent indicator that is specifically expressed in the gastrointestinal tract and related to the IBD pathway. In the revised version, we have added a comprehensive description detailing the rationale behind our selection of SPINK4. (p.4, para.1; line 67–70)

Reference

1. Huang, B., *et al.* Mucosal Profiling of Pediatric-Onset Colitis and IBD

Reveals Common Pathogenics and Therapeutic Pathways. *Cell* **179**, 1160-1176.e1124 (2019).

Comment 3: Is the recombinant SPINK4 a human protein? Is it mouse protein? There is no description.

Response 3: Both human- and mouse-derived recombinant SPINK4 proteins were enrolled. The colitis model mice were treated with the mouse-derived SPINK4 protein, whereas the *ex vivo* experiment was conducted with recombinant human SPINK4. We have provided conclusive information on SPINK4 in the Methods section. (line 105, 402, 514, 516, 529, 660)

Comment 4: Did Conditional *Spink4* KO mice have any phenotype?

Response 4: Thank you for your enlightening question. The conditional *Spink4* KO mice showed no disease phenotypes under specific-pathogen-free conditions. The length and appearance of the small intestine and colon showed no significant difference from those of the wild type. Upon histological evaluation, there was no evidence of inflammation infiltration in the intestine. We have added the phenotype description of *Spink4*-conditional knockout mice to the revised text. (p.6, para.2; line 118–122)

Comment 5: The Murine Gene symbol of “SPINK4” should be “*Spink4*”.

Response 5: Thanks for the valuable suggestion. We have replaced “*SPINK4*” with

“*Spink4*” when referring to the murine gene.

Comment 6: On page 7 line 150, “SPINK family members share a similar Kazal motif like EGF”. The authors should show the reference

Response 6: As suggested, we have added the reference to the revised manuscript (ref. 22)

Reviewer #2 (Remarks to the Author):

This experimental study deals with the role the secreted protease inhibitor SPINK4 may have in colitis, and especially in IBDs. The Authors described and proved its relationship with IBD in DSS- and TNBS-colitis mouse models and in humans, and linked their serum concentrations with disease activity, with the highest levels of the molecule found in the remission stage of the disease. Moreover, they evaluated the therapeutic potential of the administration of recombinant SPINK4 in mice. It proved to be effective in reducing intestine inflammation per se and even having a synergistic effect when combined to anti-TNF α drugs. Most of the effect of SPINK4 is due to the activation of the EGFR pathway and the consequent induction of goblet cell differentiation. This enhances the mucus production, proved by the direct correlation with the increase of the MUC2 expression. Also, SPINK4 levels seem to be positively related to Pam2CSK4 ones, whose action on the other hand mimics the one of Gram negative bacteria LPS.

The amount of data brought by the Authors regarding this molecule made light on a new etiopathogenetical mechanism in IBDs, and potentially on a possible therapeutic application.

I ask the Authors to better clarify or explain the following points:

Comment 1: • 40-41: “The role of colonic GCs, typically characterized by mucin production, in colitis has been ignored”. Considering after this statement references regarding GC in IBD follow, especially number 15 and 19, I’d modify the “has been ignored” with “has been underestimated” or similar.

Response 1: Thank you for your kind suggestion. The term “ignored” has been replaced with “underestimated” in the revised manuscript. (line 40)

Comment 2: • 80-81: “however, SPINK4 was barely expressed in the active stage”. So was this difference statistically significant when compared to controls or not?

Response 2: The p value between the groups of active stage and controls was 0.0031 and the difference was statistically significant. We have clarified this in the revised text. (p.4, para.3; line 83–84)

Comment 3: • 105-106: “This group had ... less reduced intestinal permeability”. I’d modify “less reduced” with a simpler “increased”, if so.

Response 3: Thanks for your advice. However, the tendency of intestinal permeability after treatment of rSPINK4 remained, and the difference was not statistically significant. We would like to modify the description with “marginally reduced”. (line

108)

Comment 4: • 107-109: “we further tested GCs numbers and the mucus thickness and found that both of them in the rSPINK4 treatment group were obviously increased”. I’d suggest to remove the “obviously”

Response 4: As suggested, we have removed “obviously” in the revised version of manuscript. (line 111)

Comment 5: • 226-228: “Therefore, we cocultured LS174T, a colonic cancer cell line, which mimics GCs in vitro on account of abundant mucous vesicles, with microbiotic elements”. A quote is necessary.

Response 5: The related reference has been added to illustrate the background of LS174T in the Results section. (ref. 18, 30, 31)

Comment 6: • 228-229: “Pam2CSK4, a synthetic diacylated lipopeptide that activates the TLR2/TLR6/229 heterodimer”. A quote is necessary.

Response 6: Thanks for your suggestion. A reference has been added in support of the description of Pam2CSK4. (ref. 32)

Comment 7: • 231-232: “The other microbiota components”. The same reference I suggested for 226-228 should be quoted here too, to clarify which microbiota components the Authors are referring to.

Response 7: The reference for the microbiota components has been added to the revised manuscript. (ref. 18)

Comment 8: • 245-246: “we estimated the mucus permeability, which is the most important intrinsic property of GCs”. Mucus permeability is not a property of GC, mucus production is.

Response 8: Thank you for pointing this out. The term “GC” has been revised to “mucus” to imply our intended meaning. (line 274)

Comment 9: • 245-255: “Moreover, there was a mild difference between the remission and active stages in CD patients”. I suggest to explicit in the text if this difference was statistically significant or not.

Response 9: Thanks for your careful review. The difference in serum SPINK4 levels between remission and active stages in CD patients, as determined by clinical assessment, was statistically significant. The expression in the active group was 1.5-fold higher than that in the remission group. Therefore, “with statistical difference” was added to the revised manuscript. (line 283)

Comment 10: • 280-282: “The inability of IBD to be self-healing may be attributed to... other pathogenetic mechanisms”. A quote is necessary.

Response 10: Relevant reference citations to support the text on other pathogenetic mechanisms of IBD have been included in the revised text. (ref. 37, 38)

Comment 11: • 283-284: “Though numerous serological markers burst out”. A quote is necessary.

Response 11: The relevant references have been included in the revised text. (ref. 39, 40)

Comment 12: • 314: “which kind of microbiota”. I suggest to modify it in “which microbial composition” or “microbial components”.

Response 12: Thank you for your thoughtful advice. We have modified the term “which kind of microbiota” to “which microbial components” in the revised manuscript. (line 346)

Comment 13: • 334-337: “Five UC patients and four age- and sex-matched healthy controls were enrolled. The inflamed region at the sigmoid and uninflamed region at the ascending colon respectively served as UC and SC groups.” Why did the Authors decided to consider different regions for the two groups? Then, with “SC” do the Authors mean Healty Controls (HC)?; in case, the acronym must be changed.

Response 13: We apologize for the confusion. Three groups were enrolled in this study for scRNA-seq test, including UC, SC and HC. The “UC” denotes the inflamed region from patients with UC, while the “HC” represents the uninflamed region from healthy control individuals. The abbreviation “SC” in the original text refers to “self-control”, which means the uninflamed colon of the same UC patients. Considering the

retrograde involvement of intestinal inflammation in UC patients, the inflamed region at the sigmoid and uninflamed region at the ascending colon were respectively identified as the UC and SC groups. HC samples from the sigmoid in healthy controls were also enrolled in the scRNA-seq test. We have modified the sentences to better distinguish between the different groups, such as self-control (SC) and healthy control (HC). (p.18, para.2; line 370–379)

Comment 14: • 337: “Biopsy specimens from healthy controls served as the HC group”: where the biopsies taken from the segments described in 336? I suggest to explicit it.

Response 14: As shown in Comment 13, the intestinal biopsies were collected from the sigmoid colon in the HC group, which is more comparable to the UC group. As suggested, we have provided a more detailed description of the HC group in the revised text. (p.18, para.2; line 370–379)

Reviewer #3 (Remarks to the Author):

This manuscript is focused on the regulatory effects of serine protease inhibitor Kazal-type 4 (SPINK4) on intestinal inflammation. The conclusions provided by the authors include: SPINK4 is associated with the colitis status in IBD patients; the increase in colitis induced by DSS and TNBS in mice with deletion of SPINK4 is rescued by recombinant SPINK4, which is synergistic with an TNF inhibitor,

infliximab; SPINK4 promotes goblet cell differentiation through a Kazal-like motif-modulated EGFR-Wnt/ β -catenin and -Hippo pathways; microbiota-derived diacylated lipoprotein Pam2CSK4 triggers SPINK4 production. The author also stated that SPINK4 in circulation could serve as a standard to assess disease status in patients with IBD.

Although large amount of data is shown in this manuscript, useful data to support the conclusion is lacking.

Comment 1: More data from mice with EGF receptor specifically deleted in intestinal epithelial cells and EGF receptor knock out goblet cell lines are needed to support the involvement of EGF receptor signaling in SPINK4-regulated cellular responses.

Response 1: Thanks for your constructive advice. Unfortunately, there were no readily available EGF receptor conditional knockout mice. Establishment of these conditional knockout mice require a significant investment of both time and financial resources. We have addressed this limitation in the Discussion section (p.17, para.1; line 351–352).

Nonetheless, we have successfully constructed an EGF receptor knockout goblet cell line as suggested. The validation of SPINK4-regulated cellular responses was conducted in this stable clone. The new data have been added in the revised manuscript (Figure S5, S7; p.8, para.3; p.11, para.4; p.20, para.2; line 172–173, 238–239, 414–424)

Comment 2: The interaction of EGF receptor and SPINK4 identified by the co-immunoprecipitation and pull-down assays is not the strong evidence to support SPINK4 binding to the ligand binding site on EGF receptor to activate EGF receptor. EGF receptor ligand competition assay may support some evidence whether SPINK4 could function as a EGF receptor ligand. Thus, more studies are needed to elucidate the mechanisms by which SPINK4 activates EGF receptor.

Response 2: We would like to appreciate for your suggestion. In the revised version, we have validated the interaction between SPINK4 and EGF receptor using an ITC test and colocalization analysis, as well as through original co-immunoprecipitation and pull-down assays. The recombinant EGF protein has been used in the pull-down binding test between rSPINK4 and rEGFR as the classical competition assay described in the previous study ¹. The AlphaLISA EGF and EGFR binding kits ² were used to quantitatively evaluate SPINK4 binding activity targeting EGFR, which was compared to EGF. The detailed methods and results have been presented in the revised manuscript. (Figure 4; p.11, para.2; p.25, para.2; p.26, para.4; line 225–227, 520–523, 552–562)

Reference

1. Zheng, L.S., *et al.* SPINK6 Promotes Metastasis of Nasopharyngeal Carcinoma via Binding and Activation of Epithelial Growth Factor Receptor. *Cancer research* **77**, 579-589 (2017).
2. Ticiani, E., *et al.* Bisphenol S and Epidermal Growth Factor Receptor

Signaling in Human Placental Cytotrophoblasts. *Environ. Health Perspect.*
129, 27005 (2021).

Comment 3: There are no studies to investigate the effects of Pam2CSK4 on colitis and mucus production in vivo.

Response 3: Thanks for your suggestion. To further elucidate the function of Pam2CSK4, we established a DSS-induced colitis model administrated by Pam2CSK4. Consistent with our expectation, Pam2CSK4 intervention in vivo alleviated colitis and stimulated the production of SPINK4 and mucin. The detailed methods and results have been provided in the revised version of the manuscript. (Figure S8; p.12, para.2; p.31, para.3; line 261–267, 666–667)

Comment 4: In the Method section, the authors stated “Villin-Cre mice were obtained from the Jackson Laboratory (USA) and the SPINK4 floxed mice were generated by GemPharmatech Co., Ltd, based on the designed sgRNAs”. No mouse background was provided. No experimental data to confirm the specific deletion of SPINK4 in intestinal epithelial cells in mice. No information about the phenotype of this transgenic mice. The authors defined these mice as “conditional knockout mice” in this manuscript. Is the gene mutation in floxed SPINK4-Villin-Cre mice conditional or constitutive?

Response 4: Thanks for your comment. Based on the expression pattern of Villin, the SPINK4 gene is conditionally deleted in the digestive tract's epithelium in *Spink4-*

Villin-Cre mice. In the revised Methods section, we have provided a comprehensive description of the background and detailed information on these knockout mice. (p.30, para.1; line 626–628)

Additionally, the specific deletion of SPINK4 from the intestinal epithelium has been confirmed using histological test and further elaborated upon in the supplementary material (Supplementary Fig. 3H). As mentioned in our response to Comment 4 from Reviewer 1, no evidence of enteric inflammation under normal condition was detected either macroscopically or microscopically. We have added a description of phenotype of SPINK-conditional knockout mice to the revised text. (p.6, para.2; line 117–122)

Comment 5: In the result section, only conclusion from each figure is provided. The authors should present detailed data and explanation to support the conclusion.

Response 5: Thanks for your careful evaluation. Detailed description and explanation of figures have been presented in the Results section. (p.4, para.3; p.5, para.1; p.6, para.1-2; p.7, para.2; p.8, para.1; p.9, para.2; p.10, para.2; p.11, para.4; p.12, para.2)

Comment 6: No statement about how many times of the experiments are repeated for *in vitro* studies.

Response 6: We have repeated the experiments at least three times and highlighted this point in the Methods section. (line 705)

Comment 7: Higher magnification images of HE stained colonic sections are needed to examine the histological changes (Figure 2O).

Response 7: We would like to retain the original HE images from the Swiss roll intestine, which is an overview of the inflamed intestine. However, we have added the representative inflamed part at a higher magnification for comparison in Figure S4O simultaneously. (Figure S4O)

Comment 8: It is difficult to get any useful information from endoscopic images (Figure 2K and Figure 3F).

Response 8: Thanks for your question. We have added a detailed description of the endoscopic images to the Results section and figure legends. (p.6, para.2; p.7, para.2; p.43, para.1; p.46, para.1; line 120–122, 145–147, 906–907, 932–933)

Comment 9: It is not clear whether colonic epithelial cells or total colonic tissues are used for the Western blot analysis of Muc 2 levels (Figure 2J) and EGF receptor phosphorylation (Figure 3H). The molecular weight of Muc2 is identified as around 130kDa in this manuscript. The authors need to confirm it.

Response 9: Thank you for your careful review. The localization of MUC2 and phosphorylated EGFR was confirmed via histological assessment, revealing predominant expression in the intestinal epithelium for both proteins. We then used the whole colonic tissues to confirm the levels of MUC2 and phosphorylated EGFR by western blot test.

With regard to MUC2, the molecular weight of whole MUC2 was determined to be 540 kDa. The MUC2 antibody, which was used in western blot analysis, was purchased from Abcam (ab272692). The observed band size was approximately 140-170 kDa, consistent with the size indicated in the manufacturer's instructions and related reference ^{1,2}. The detailed website URL is provided as follows:

<https://www.abcam.cn/products/primary-antibodies/muc2-antibody-epr23479-47-ab272692.html>

Reference

1. Ruan, Z., Yu, Y., Han, P., Zhang, L. & Hu, Z. Si-Wu Water Extracts Protect against Colonic Mucus Barrier Damage by Regulating Muc2 Mucin Expression in Mice Fed a High-Fat Diet. *Foods (Basel, Switzerland)* **11**(2022).
2. Zhan, Y., *et al.* Effects of Maren Pills on the Intestinal Microflora and Short-Chain Fatty Acid Profile in Drug-Induced Slow Transit Constipation Model Rats. *Front. Pharmacol.* **13**, 804723 (2022).

Comment 10: Fold change of band density in Western blot from all in vitro experiments and mice should be provided.

Response 10: Thank you for your suggestion. We have performed band density analysis for all western blot tests using Image-Pro plus in figures or source data and detailed methods have been provided in the revised manuscript. (p.23, para.1; line 475–476)

REVIEWERS' COMMENTS

Reviewer #1 (Remarks to the Author):

The manuscript was revised appropriately.

Reviewer #2 (Remarks to the Author):

I was satisfied with the corrections done by the Authors. The answers to my points were valid.

Reviewer #3 (Remarks to the Author):

Authors addressed most critiques very well. The revised manuscript is significantly improved. However, regarding Reviewer 3's Comment 1 (More data from mice with EGF receptor specifically deleted in intestinal epithelial cells and EGF receptor knock out goblet cell lines are needed to support the involvement of EGF receptor signaling in SPINK4-regulated cellular responses), authors provided improper responses. Authors stated that "there were no readily available EGF receptor conditional knockout mice. Establishment of these conditional knockout mice require a significant investment of both time and financial resources".

Egfrfl/fl mice have been used by several groups in published papers. Mice with EGF receptor specifically deleted in intestinal epithelial cells can be obtained by crossing Egfrfl/fl mice with Villin-cre mice.

Response Letter to Reviewers:

We sincerely thank the editors and reviewer team for your consideration on publication of our work. We have thoroughly revised the comments from each reviewer and submitted necessary files, including the latest version of manuscript, the response letter, the resubmission cover letter and other supplementary materials listed in the Author Checklist. Additionally, we have ensured that the figures and tables comply with the guidelines. In the file, comments are indicated in blue and our responses are presented in black. Please see below out point-by-point responses.

Reviewer #1 (Remarks to the Author):

The manuscript was revised appropriately.

I am delighted to hear that you are satisfied with our revisions. Thank you for your valuable feedback on this project.

Reviewer #2 (Remarks to the Author):

I was satisfied with the corrections done by the Authors. The answers to my points were valid.

We appreciate for your careful review of the manuscript. Thanks a lot for your support and recognition of our work.

Reviewer #3 (Remarks to the Author):

Authors addressed most critiques very well. The revised manuscript is significantly improved. However, regarding Reviewer 3's Comment 1 (More data from mice with EGF receptor specifically deleted in intestinal epithelial cells and EGF receptor knock out goblet cell lines are needed to support the involvement of EGF receptor signaling in SPINK4-regulated cellular responses), authors provided improper responses. Authors stated that "there were no readily available EGF receptor conditional knockout mice. Establishment of these conditional knockout mice require a significant investment of both time and financial resources".

Egfrfl/fl mice have been used by several groups in published papers. Mice with EGF receptor specifically deleted in intestinal epithelial cells can be obtained by crossing

Egfrfl/fl mice with Villin-cre mice.

Thanks for your constructive advice and comment on our revision. We acknowledge that the financial and time restrictions prevented us from including EGF receptor conditional knockout mice in our study. Instead, we used EGF receptor knockout goblet cell lines as supplementary information. The utilization of EGF receptor knockout mice will be considered in our future experiments for further validation, if they are accessible.